# Hair follicle epidermal stem cells define a niche for tactile sensation

Chun-Chun Cheng[1†‡], Ko Tsutsui[1†], Toru Taguchi[2,3†], Noriko Sanzen[1], Asako Nakagawa[1], Kisa Kakiguchi[4], Shigenobu Yonemura[4,5], Chiharu Tanegashima[6], Sean D Keeley[7§], Hiroshi Kiyonari[8], Yasuhide Furuta[8], Yasuko Tomono[9], Fiona M Watt[10], Hironobu Fujiwara[1*]

[1]Laboratory for Tissue Microenvironment, RIKEN Center for Biosystems Dynamics Research, Kobe, Japan; [2]Department of Neuroscience II, Research Institute of Environmental Medicine, Nagoya University, Nagoya, Japan; [3]Department of Physical Therapy, Niigata University of Health and Welfare, Niigata , Japan; [4]Laboratory for Ultrastructural Research, RIKEN Center for Biosystems Dynamics Research, Kobe, Japan; [5]Department of Cell Biology, Tokushima University Graduate School of Medical Science, Tokushima, Japan; [6]Laboratory for Phyloinformatics, RIKEN Center for Biosystems Dynamics Research, Kobe, Japan; [7]Phyloinformatics Unit, RIKEN Center for Life Science Technologies, Kobe, Japan; [8]Laboratories for Animal Resource Development and Genetic Engineering, RIKEN Center for Biosystems Dynamics Research, Kobe, Japan; [9]Division of Molecular and Cell Biology, Shigei Medical Research Institute, Okayama, Japan; [10]Centre for Stem Cells and Regenerative Medicine, King's College London, London, United Kingdom

*For correspondence:
hironobu.fujiwara@riken.jp

[†]These authors contributed equally to this work

Present address: [‡]Simmons Comprehensive Cancer Center, UT Southwestern Medical Center, Texas, United States; [§]Deutsche Forschungsgemeinschaft (DFG) Center for Regenerative Therapies, Technische Universität Dresden, Dresden, Germany

Competing interests: The authors declare that no competing interests exist.

**Abstract** The heterogeneity and compartmentalization of stem cells is a common principle in many epithelia, and is known to function in epithelial maintenance, but its other physiological roles remain elusive. Here we show transcriptional and anatomical contributions of compartmentalized epidermal stem cells in tactile sensory unit formation in the mouse hair follicle. Epidermal stem cells in the follicle upper-bulge, where mechanosensory lanceolate complexes innervate, express a unique set of extracellular matrix (ECM) and neurogenesis-related genes. These epidermal stem cells deposit an ECM protein called EGFL6 into the collar matrix, a novel ECM that tightly ensheathes lanceolate complexes. EGFL6 is required for the proper patterning, touch responses, and αv integrin-enrichment of lanceolate complexes. By maintaining a quiescent original epidermal stem cell niche, the old bulge, epidermal stem cells provide anatomically stable follicle–lanceolate complex interfaces, irrespective of the stage of follicle regeneration cycle. Thus, compartmentalized epidermal stem cells provide a niche linking the hair follicle and the nervous system throughout the hair cycle.
DOI: https://doi.org/10.7554/eLife.38883.001

## Introduction

Tissue stem cells in many epithelia, including the epidermis, intestine, mammary glands and lungs, often comprise heterogeneous populations with distinct transcriptional and anatomical features (*Donati and Watt, 2015*; *Xin et al., 2016*). Individual stem cell pools contribute to tissue maintenance through different homeostatic and regenerative properties and give rise to distinct epithelial compartments, while they also hold plasticity to change their identity and function when perturbed. These recent findings question the traditional view of the role of stem cell heterogeneity, which underlies the theory of unidirectional stem cell differentiation hierarchy (*Goodell et al., 2015*).

Although the biological significance of the heterogeneity and compartmentalization of epithelial stem cells has primarily been studied in the context of epithelial tissue maintenance and regeneration, its other physiological roles remain unclear.

Hair follicles are highly conserved touch sensory organs in the hairy skin that covers most of the mammalian body surface and detects touch signals essential for development and survival (*Lumpkin et al., 2010*). Hair follicles are innervated by mechanosensory end organs called lanceolate complexes, which are composed of parallel, longitudinally aligned low-threshold mechanoreceptor (LTMR) axonal endings and terminal Schwann cell processes (*Zimmerman et al., 2014*). Hair follicles are also connected to arrector pili muscles (APMs), thus forming a unique regenerating motile sensory organ. These two hair follicle appendages attach to the permanent portion of the hair follicle, known as the bulge, where epidermal stem cells reside.

Hair follicle epidermal stem cells are heterogeneous in their molecular and functional properties and compartmentalized along the longitudinal axis of the hair follicle (*Figure 1A*) (*Solanas and Benitah, 2013*). The mid-bulge region contains $CD34^+$ slow-cycling epidermal stem cells that serve as a reservoir of epidermal stem cells and were once thought to be the master stem cells at the top of the epidermal hierarchy. A series of recent studies, however, identify several additional pools of different epidermal stem cells around the bulge (e.g. in the isthmus, junctional zone, upper-bulge and hair germ). For example, the upper-bulge is a niche for $Gli1^+$ epidermal stem cells and they contribute to hair follicle regeneration and wound healing of interfollicular epidermis (*Brownell et al., 2011*). The importance of epidermal stem cell heterogeneity and compartmentalization has almost entirely been explained by their role in regional tissue replenishment to maintain different functional compartments of the epidermis, but its roles apart from epidermal maintenance remain poorly understood. As coordinated epithelial–mesenchymal interactions are essential for hair follicle development, regeneration and functioning, the unique signaling territories and tissue architecture provided by compartmentalized epidermal stem cells probably regulate the interactions between the epidermis and a variety of hair follicle–associated structures, including sensory nerves, APMs and the dermal papilla. Consistent with this idea, it has been reported that mid-bulge epidermal stem cells create a specialized basement membrane containing nephronectin, thereby providing a niche for APM development and anchorage (*Fujiwara et al., 2011*). Follicle epidermis–derived BDNF is also critical for follicle-nerve interactions (*Rutlin et al., 2014*).

In this study, we set out to address the idea that upper-bulge epidermal stem cells are molecularly and anatomically specialized for interaction with lanceolate complexes and thus involved in generating the sense of touch. We demonstrated that upper-bulge epidermal stem cells produce a special extracellular matrix (ECM) environment and unique epidermal tissue architecture that creates a stable hair follicle–lanceolate complex interface for tactile sensation. Thus, epidermal stem cell heterogeneity and compartmentalization appear to be important for both regional epidermal maintenance and patterned epithelial–mesenchymal interactions.

## Results

### Upper-bulge epidermal stem cells are molecularly specialized for hair follicle–nerve interactions

We first examined the global transcriptional features of distinct epidermal stem cell populations in the hair follicle. To this end, we established FACS-based cell purification methods using several eGFP reporter mouse lines that label different stem cell compartments. This way, we isolated cellular subpopulations resident in the lower-isthmus ($Lgr6^+$), upper-bulge ($Gli1^+$), mid-bulge ($CD34^+$), and hair germ ($Cdh3^+$), as well as unfractionated basal epidermal stem cells ($\alpha6$ integrin$^+$) (*Figure 1A and B*, *Figure 1—figure supplement 1A*). The purity of the isolated populations was verified by the expression of compartment-specific genes (*Figure 1—figure supplement 1B and C*, Materials and methods) and we performed RNA-seq on these isolated cell populations. Gene Set Enrichment Analysis (GSEA) indicated that neurogenesis-related Gene Ontology (GO) terms are over-represented in $Gli1^+$ bulge epidermal stem cells (*Figure 1C*, *Figure 1—source data 2*). To further identify $Gli1^+$ compartment–enriched genes, we performed a pairwise transcriptional comparison between the $Gli1^+$ population and all the other populations and plotted the relationship between $Gli1^+$-enriched gene sets in a Venn diagram (*Figure 1D*). Genes categorized in 'Group I' are $Gli1^+$ enriched genes.

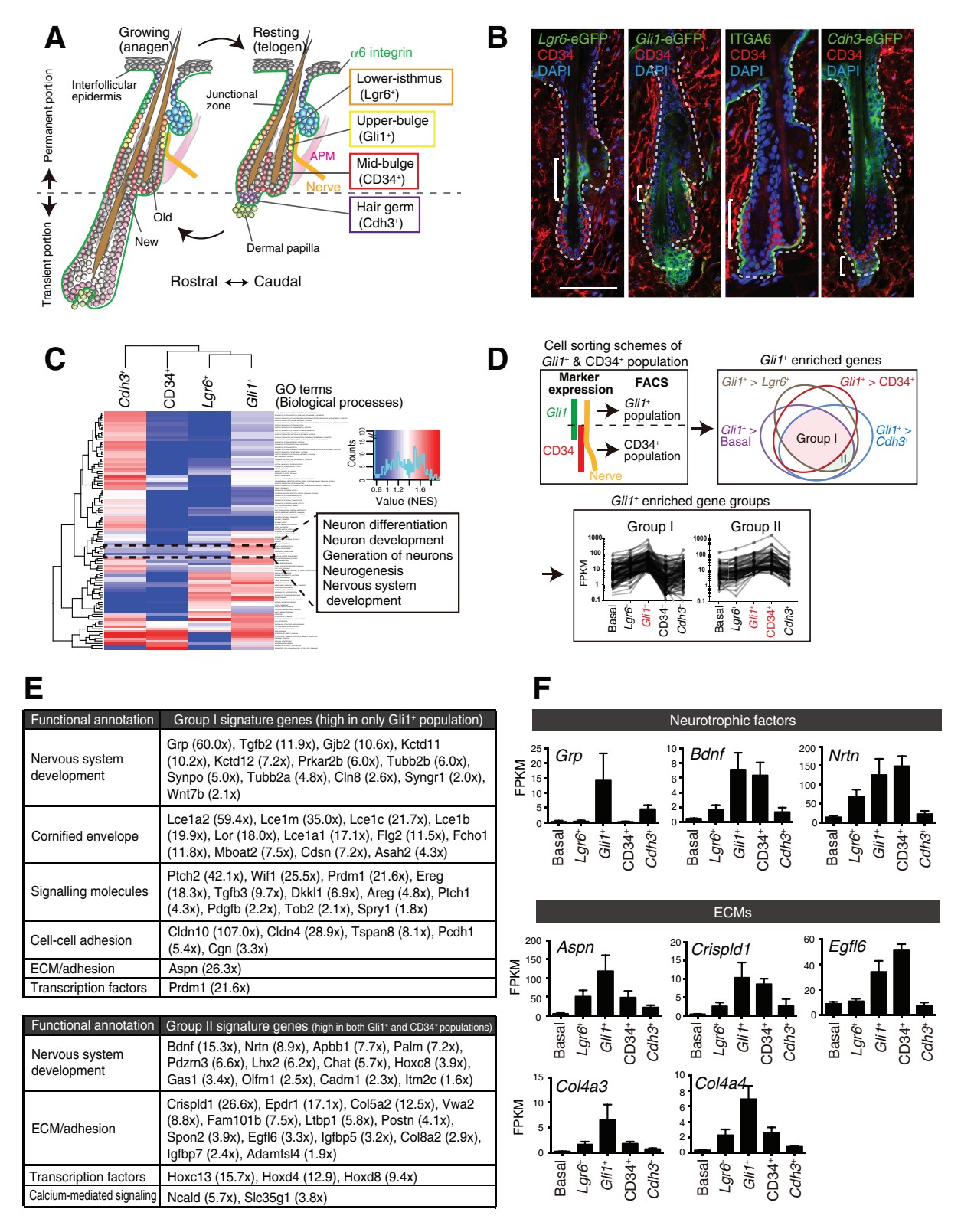

**Figure 1.** Upper-bulge epidermal stem cells are molecularly specialized for hair follicle–nerve interactions. (**A**) Graphical illustration of epidermal stem cell compartments. APM, arrector pili muscle. (**B**) Distinct epidermal stem cell compartments in 8-week-old telogen skin were visualized with specific eGFP reporters and cell surface markers. Brackets indicate target cell populations for sorting. (**C**) Z-score heat map representing the normalized enrichment score (NES) of GSEA using the transcriptome data of each epidermal stem cell population. (**D**) Scheme of the extraction of

*Figure 1 continued on next page*

*Figure 1 continued*

genes highly expressed in the *Gli1*$^+$ population (Group I) and in both the *Gli1*$^+$ and CD34$^+$ populations (Group II). (**E**) Lists of the genes highly expressed in *Gli1*$^+$ population (Group I) and in both *Gli1*$^+$ and CD34$^+$ populations (Group II). (**F**) Expression levels of neurotrophic factor and ECM genes highly expressed in upper-bulge stem cells. Basal, basal epidermal stem cell pool. Data are mean ±SD, n = 3–4.
DOI: https://doi.org/10.7554/eLife.38883.002

The following source data and figure supplement are available for figure 1:

**Source data 1.** Raw numerical data for *Figure 1* and associated figure supplements.
DOI: https://doi.org/10.7554/eLife.38883.004
**Source data 2.** Normalized Enrichment Score (NES) of Gene Set.
DOI: https://doi.org/10.7554/eLife.38883.005
**Figure supplement 1.** FACS-based cell isolation procedures for distinct epidermal stem cell populations.
DOI: https://doi.org/10.7554/eLife.38883.003

We also extracted genes included in 'Group II', which are genes highly expressed both in the *Gli1*$^+$ population and the CD34$^+$ population, since *Gli1* and CD34 double-positive cells were included in the CD34$^+$ population in our sorting scheme (*Figure 1D*). Prominent gene-annotation clusters in both Group I and Group II cells encode proteins involved in nervous system development, including the neurotrophic factors *Grp*, *Bdnf* and *Nrtn* and the keratitis-ichthyosis-deafness syndrome gene *Gjb2* (*Figure 1E and F*). Multiple ECM genes are also upregulated in the upper-bulge compartment, including *Aspn*, *Crispld1*, *Egfl6*, and the deafness-related ECM genes *Col4a3* and *Col4a4* (*Mochizuki et al., 1994*) (*Figure 1E and F*). This global gene expression profiling of compartmentalized epidermal stem cells suggests that upper-bulge epidermal stem cells are specialized both to interact with the nervous system and to express a unique set of ECM genes.

## Upper-bulge epidermal stem cells deposit EGFL6 into the collar matrix

It has been suggested that the ECM plays important roles in mammalian touch end organs, but the molecular identity and functions of this putative ultrastructure remain unknown (*Lumpkin et al., 2010*; *Zimmerman et al., 2014*). On examining the tissue localization of 15 upper-bulge ECM proteins, we found that 8 ECM proteins were deposited in the upper-bulge (*Figure 2A*, *Figure 2— source data 2*). Among them, EGFL6 (EGF-like domain multiple 6) exhibited the most restricted localization in the upper-bulge of all types of dorsal hair follicles and showed a unique C-shaped pattern with a gap at the rostral side of the hair follicle (*Figure 2B*). βIII-tubulin staining showed that skin nerve endings terminate at the EGFL6 deposition sites (*Figure 2B*). Magnified 3D images revealed the close association of EGFL6 with longitudinal lanceolate parallel LTMR axonal endings of lanceolate complexes, which are activated by tactile stimuli (*Figure 2C*) (*Bai et al., 2015*), and longitudinal processes of nestin-positive non-myelinating terminal Schwann cells of lanceolate complexes (*Figure 2D*). A close anatomical association of follicle EGFL6 with blood vessels has been reported (*Xiao et al., 2013*), but their direct contact was not observed (*Figure 2E*). To detect cells expressing *Egfl6* mRNA, we generated *Egfl6-H2b-Egfp* reporter mice. eGFP protein expression was enriched in upper-bulge epidermal stem cells at the caudal side of the hair follicle, but was not detected in dermal cells around the upper-bulge or dorsal root ganglia neurons that innervate the upper-bulge (*Figure 2F and G*). Although EGFL6 expression and localization in mouse skin has been reported previously (*Osada et al., 2005*), the upper-bulge deposition of EGFL6 has not previously been identified.

We further examined EGFL6 deposition and lanceolate complex formation during postnatal skin development. At P5, disarrayed nerve fibers were detected at the upper-bulge where EGFL6 was broadly distributed (*Figure 2—figure supplement 1*). At P9, nestin-positive terminal Schwann cells appeared at the upper-bulge with disorganized cellular morphologies. After the maturation of the hair follicle (P14), sensory nerves and terminal Schwann cells formed parallel stripes, but EGFL6 showed a circumferential distribution confined to the base of lanceolate endings. At P35, the mature distribution of EGFL6 and lanceolate complexes was observed.

To investigate the ultrastructural localization of EGFL6, we performed electron microscopic analysis of transverse sections of the upper-bulge. As described by *Li and Ginty, 2014*, each LTMR axonal ending was sandwiched by processes of terminal Schwann cells (*Figure 2H*). We identified an electron-dense amorphous ECM structure surrounding the lanceolate complexes (black arrowheads in

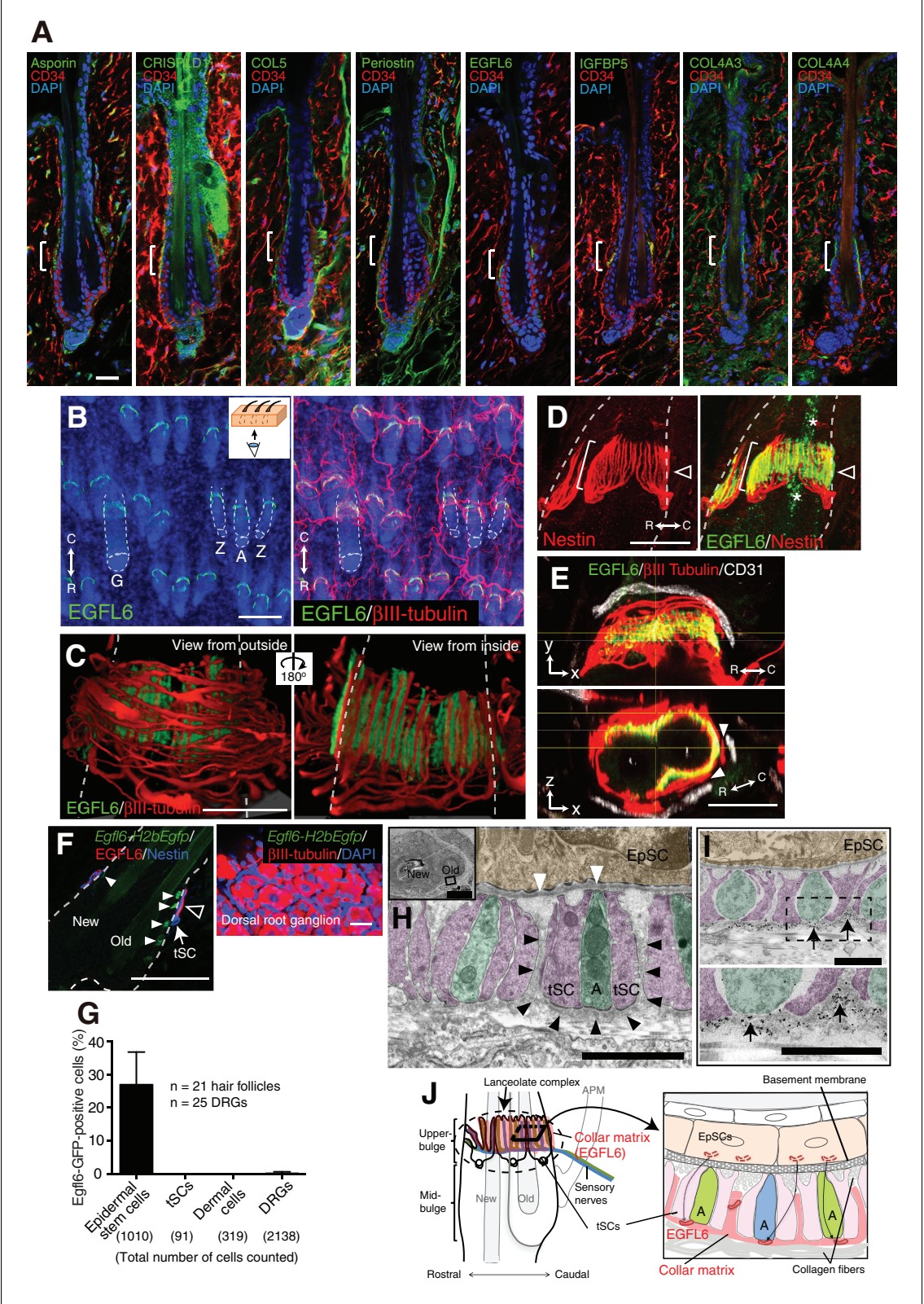

**Figure 2.** Upper-bulge epidermal stem cells deposit EGFL6 into the collar matrix. (**A**) Immunostaining pattern of upper-bulge-specific ECM in 8-week-old telogen skin. Brackets: upper-bulge. (**B**) EGFL6 was colocalized with βIII-tubulin⁺ nerve endings in the upper-bulge of dorsal guard ('G'), awl/auchene ('A') and zigzag ('Z') hair follicles. (**C**) Magnified images of EGFL6 and βIII-tubulin in telogen skin. (**D**) The protrusions of the terminal Schwann cells (white brackets) were colocalized with EGFL6 in the upper-bulge of 8-week-old telogen hair follicle (open arrowheads). Asterisks: *Figure 2 continued on next page*

*Figure 2 continued*

nonspecific signals. (**E**) CD31$^+$ blood vessels did not show direct contact with EGFL6 in 7-week-old telogen hair follicle. Arrowheads: gaps between EGFL6 (green) and blood vessels (white). (**F**) The dorsal skin and dorsal root ganglia (DRG) of an 8-week-old *Egfl6-H2b-Egfp* mouse was stained for eGFP, EGFL6, nestin and βIII-tubulin. eGFP (closed arrowheads) staining was observed in the upper-bulge (open arrowhead) epidermis of old bulge (Old), but not in other cellular components around the upper-bulge. (**G**) Statistical examination of *Egfl6*-H2BeGFP-positive cells around the upper bulge. tSC, termimal Schwann cell. Data are mean ± SD. (**H**) Transmission electron microscopic image of the upper-bulge of P35 zigzag hair follicle showing an electron-dense amorphous ECM structure, the collar matrix (black arrowheads) and ensheathed lanceolate complex endings [A, axon (green); tSC, terminal Schwann cell (pink); EpSC, epidermal stem cell (gold); white arrowheads, epidermal basement membrane]. (**I**) The EGFL6-gold particles were localized in the collar matrix (arrows). (**J**) Schematic representation of the localization of EGFL6 in the upper-bulge. Scale bars, 10 μm (**A**), 20 μm (**C–F**), 100 μm (**B**), 2 μm (**H**), 1 μm (**I**).

DOI: https://doi.org/10.7554/eLife.38883.006

The following source data and figure supplement are available for figure 2:

**Source data 1.** Raw numerical data for *Figure 2*.
DOI: https://doi.org/10.7554/eLife.38883.008
**Source data 2.** List of ECM proteins screened for upper-bulge localization with immunohistochemical analysis.
DOI: https://doi.org/10.7554/eLife.38883.009
**Figure supplement 1.** EGFL6 localization and lanceolate complex formation during hair follicle development.
DOI: https://doi.org/10.7554/eLife.38883.007

*Figure 2H*). We named this novel structure the 'collar matrix', as it tightly ensheathes the lanceolate complexes. Terminal Schwann cell processes have two openings, at the follicular basement membrane and at the opposite side of the cell, thus exposing axonal endings and terminal Schwann cell processes to at least two different ECMs, the hair follicle basement membrane and the collar matrix. Although EGFL6 is expressed by the upper-bulge epidermal stem cells, an immunogold-labeled EGFL6 antibody was detected exclusively in the collar matrix (*Figure 2I*), indicating that upper-bulge epidermal stem cells secrete EGFL6, which is incorporated in the collar matrix that ensheathes lanceolate complexes (*Figure 2J*).

## EGFL6 mediates cell adhesion via αv integrins

Since integrins are key cell surface receptors for cell–ECM interactions, we next explored integrins that interact with EGFL6. First, we purified full-length mouse recombinant EGFL6 protein (*Figure 3A*) and performed cell adhesion assays to test which cell lineages adhere to EGFL6. Among nine cell lines tested, one skin fibroblast and two glial cell lines adhered to EGFL6, whereas other cell types, including keratinocytes, did not (*Figure 3B*), suggesting that although EGFL6 is derived from the upper-bulge epidermal stem cells, these stem cells are incapable of interacting with EGFL6. Instead dermal cell lineages interact and adhere to EGFL6. A point mutation in the Arg-Gly-Asp (RGD) integrin recognition sequence of EGFL6 (RGE mutant) significantly reduced its cell adhesive ability (*Figure 3C*) (*Oberauer et al., 2010*; *Osada et al., 2005*). Although α8β1 integrin has been reported to mediate cell adhesion to EGFL6 (*Osada et al., 2005*), its expression was not detected in lanceolate complexes (*Figure 3D*). Thus we further performed cell adhesion inhibition assays with a panel of antibodies that block integrin function and found that cell adhesion to EGFL6 was inhibited by the αv integrin antibody, but not by the β1 integrin antibody (*Figure 3E*). Consistent with these results, αv integrin accumulated in lanceolate complexes and colocalized with EGFL6 (*Figure 3F*). Deletion of EGFL6 decreased αv integrin and impaired its accumulation (*Figure 3F and G*). A detailed examination of integrin localization revealed that integrin αv was localized to the axonal endings of lanceolate complexes and distributed on their collar matrix side (*Figure 3H*). On the other hand, integrin β1 was enriched at the interface between hair follicles and lanceolate complex endings, where the basement membrane is located, and was also detected at the outer surface of lanceolate complex endings. Integrin α3 accumulated at the inner and outer surfaces of the axonal endings, whereas integrin α6 was preferentially localized to the inner and outer surfaces of terminal Schwann cell protrusions.

Together, these results show that EGFL6 induces cell adhesion via αv integrins that accumulate in the axonal endings of lanceolate complexes in an EGFL6-dependent manner. Distinctive distributions of ECM receptors in lanceolate complexes suggest that lanceolate complexes establish

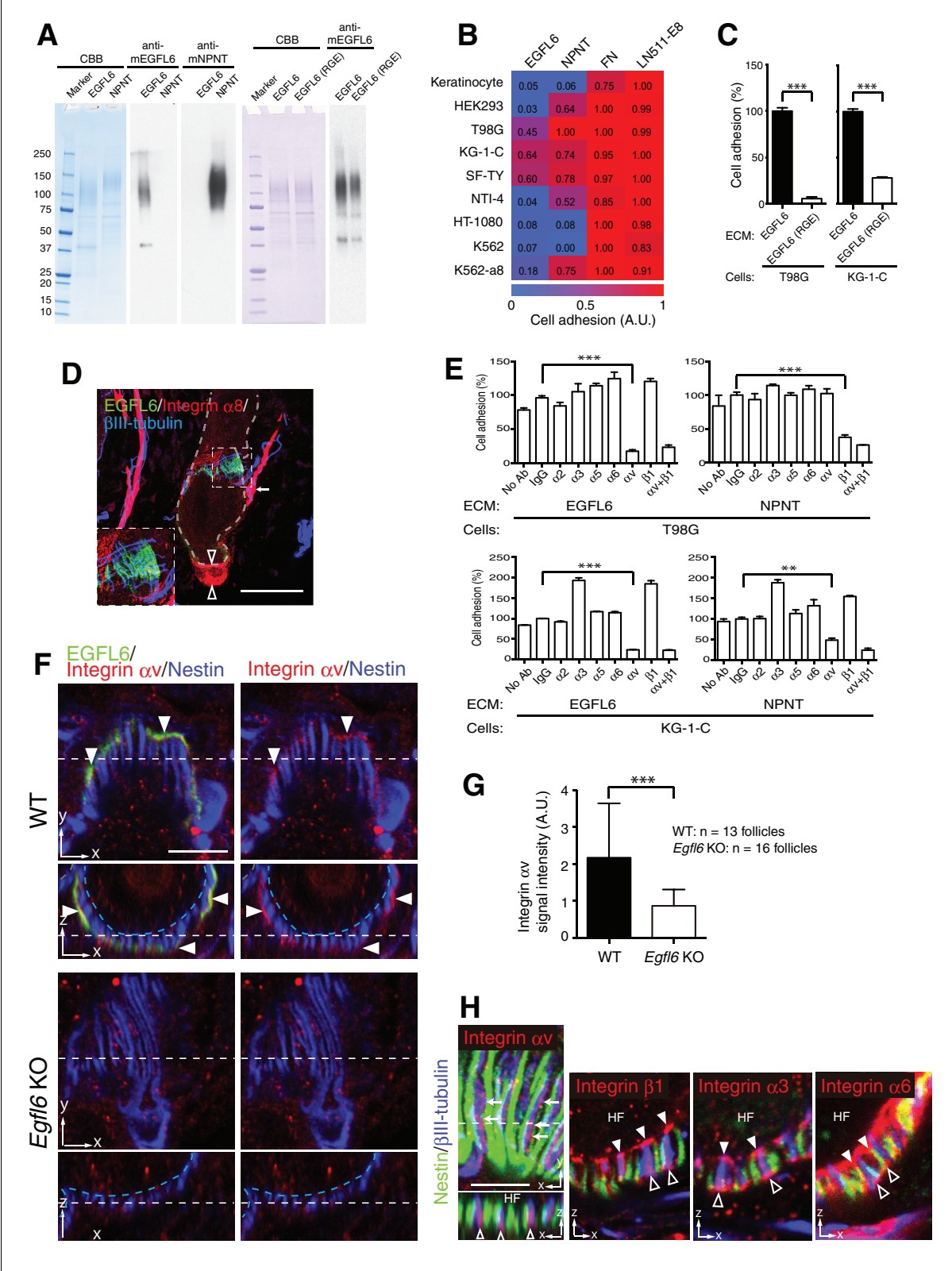

**Figure 3.** Cell adhesion to EGFL6 is mediated by αv integrins. (**A**) Coomassie Brilliant Blue (CBB) staining and western blotting of purified EGFL6, RGE-mutant EGFL6, and nephronectin (NPNT) are shown. (**B**) Heat map display of the results of cell adhesion assays. The number of adhered cells was converted to arbitrary units (A.U.). FN, fibronectin; LN511-E8, laminin-511 E8 fragment. n = 3. Cell lines used are human primary keratinocytes, human embryonic kidney epithelial cell line HEK293, human glioblastoma cell line T98G, Human glioma cell line KG-1-C, human skin fibroblast cell line SF-TY, *Figure 3 continued on next page*

*Figure 3 continued*

human embryonic fibroblast cell line NTI-4, human sarcoma cell line HT-1080, human myelogenous leukemia cell line K562 and K562 transfected with human integrin α8 cDNA K562-a8. (C) Cell adhesion assays with T98G and KG-1-C cells plated on EGFL6 or RGE-mutant EGFL6-coated dishes. Data are mean ± SEM, n = 3. (D) A maximum projection image of 8-week-old telogen dorsal hair follicles stained for EGFL6, integrin α8 and βIII-tubulin. Scale bar, 50 μm. Integrin α8 was detected in arrector pili muscles (white arrow), periphery of dermal papilla (open arrowheads) and periphery of lanceolate complexes (see the inset), but not in EGFL6-positive lanceolate complexes (see the inset). (E) Cell adhesion inhibition assays with integrin antibodies. Data are mean ± SEM, n = 3. (F) 8-week-old telogen dorsal hair follicles of wild-type and *Egfl6* knockout mice were stained for EGFL6, integrin αv and nestin. Scale bar, 10 μm. Blue dashed lines indicate the surface of follicle epidermis and white dashed lines indicate cross-section positions. (G) Quantification of αv integrin signal intensity in lanceolate complexes. Data are mean ± SD. (H) 8-week-old telogen dorsal hair follicles of wild-type mice were stained for nestin, βIII-tubulin, integrins αv, β1, α3 and α6. Sagittal view (x–y) and transverse view (x–z) images are shown. HF, hair follicle. Scale bar, 5 μm. Statistics (C, E, G): two-tailed unpaired *t*-test.

DOI: https://doi.org/10.7554/eLife.38883.010

The following source data is available for figure 3:

**Source data 1.** Raw numerical data for *Figure 3*.

DOI: https://doi.org/10.7554/eLife.38883.011

---

complex but well-organized cell–ECM interactions through interacting with the collar matrix and the basement membrane.

## EGFL6 is required for the proper patterning and touch responses of lanceolate complexes

One of the central roles of the specialized ECM around neural structures, such as the perineuronal nets, is to provide adhesive and non-adhesive physical structures to support cells and therefore determine their shape and functions (*Mouw et al., 2014*). Thus, we next examined the effect of deleting EGFL6 on lanceolate complex structures in zigzag hair follicles, which are the most common hair follicle type in rodent skin and innervated by Aδ- and C-LTMRs (*Li et al., 2011*). *Egfl6* knockout mice showed misaligned and overlapping structures of axonal endings and terminal Schwann cell processes in 8-week-old second telogen hair follicles (*Figure 4A and B*). Quantitative 3D histomorphometric analysis of LTMRs and terminal Schwann cells revealed a significant increase in overlapping points of axons and terminal Schwann cell protrusions in *Egfl6* knockout mice (*Figure 4A–C*). The numbers and lengths of axonal endings were unchanged (*Figure 4—figure supplement 1A and B*). In *Egfl6* knockout tissue, the length, but not the width, of terminal Schwann cell processes was reduced (*Figure 4—figure supplement 1C and D*). Ultrastructural analysis of lanceolate complexes identified imperfectly sandwiched lanceolate complexes in the knockout mice, but we did not detect obvious morphological abnormalities in cell–ECM and cell–cell interaction sites (*Figure 4—figure supplement 2*). We also examined the phenotype at the earlier time point, the first telogen (P19), and found that the disorganized lanceolate complex structure already occurred (*Figure 4—figure supplement 3*). *Egfl6* knockout mice did not show defects in the hair cycle (*Figure 4—figure supplement 4*). We conclude that EGFL6 is required for proper parallel patterning of lanceolate complexes.

To determine whether EGFL6 is involved in mechanotransduction, we generated skin–nerve preparations and performed a single-nerve electrophysiological analysis of Aδ-LTMRs, which are the most sensitive hair follicle–associated LTMRs, stimulated by gentle touch and hair deflection (*Bai et al., 2015*; *Rutlin et al., 2014*; *Zimmermann et al., 2009*). Although the electrical responsiveness of Aδ-LTMRs in *Egfl6* knockout skin remained intact (*Table 1*), these receptors showed clear defects in mechanical responses. The median mechanical response threshold of Aδ-LTMRs measured using von Frey hairs was significantly higher in knockout skin (*Figure 4D*). To more quantitatively analyze the mechanical sensitivity of these fibers, a light touch stimulus was applied using a piezo-controlled micromanipulator (*Figure 4—figure supplement 5A*). Averaged touch response patterns showed that knockout skin exhibits fewer action potentials than wild-type skin at displacements of >32 μm (*Figure 4E*, *Figure 4—figure supplement 5B and C*). The median threshold displacement of Aδ-LTMRs was about three times higher in the knockout (*Figure 4F*). Significantly fewer action potentials were detected during the course of the touch stimulus protocol in *Egfl6* knockout skin (*Figure 4G*). These results indicate that EGFL6 plays an important role in the excitation of Aδ-LTMRs.

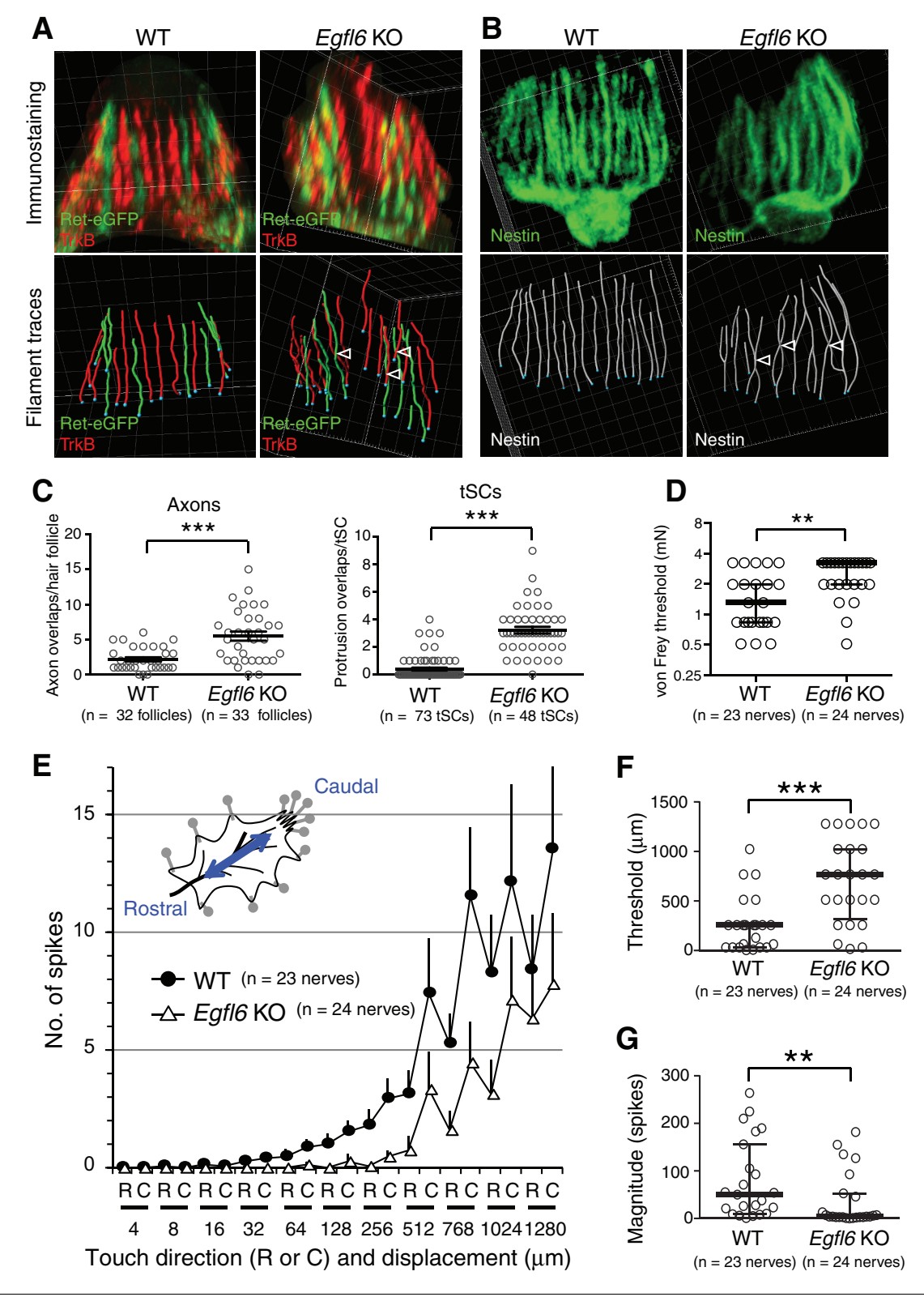

**Figure 4.** EGFL6 is required for the patterning and touch responses of lanceolate complexes. (**A and B**) 3D reconstituted images and filament tracings of axonal endings (**A**) and terminal Schwann cells (**B**) in the old bulge of wild-type and *Egfl6* knockout 8-week-old skin. *Ret*-eGFP, a marker for C-LTMRs and TrkB, a marker for Aδ-LTMRs. Open arrowhead: overlapping points of axonal endings and terminal Schwann cell processes. Grid width, 5 μm. (**C**) Overlapping points of axonal endings and terminal Schwann cell (tSC) processes were counted. Data are mean ± SEM. Two-tailed unpaired *t*-test. (**D**)

*Figure 4 continued on next page*

*Figure 4 continued*

Mechanical sensitivity of Aδ-LTMRs was analyzed using von Frey hairs in wild-type and *Egfl6* knockout skin–nerve preparations from 10 to 12 week-old telogen skin. (E) Touch response patterns of Aδ-LTMRs in wild-type and *Egfl6* knockout skin–nerve preparations. A ramp-and-hold touch stimulus was applied using a piezo-controlled micromanipulator (see Methods and *Figure 4—figure supplement 5A*). (F) Threshold displacement of Aδ-LTMRs. (G) Number of action potentials during the course of the touch stimulus protocol (i.e., magnitude of touch responses) was measured. Data are median with interquartile range (IQR) in (D, F, G) and mean ± SEM in (E). Statistics: Mann-Whitney U-test (D, F, G).
DOI: https://doi.org/10.7554/eLife.38883.012

The following source data and figure supplements are available for figure 4:

**Source data 1.** Raw numerical data for *Figure 4* and associated figure supplements.
DOI: https://doi.org/10.7554/eLife.38883.018

**Figure supplement 1.** Quantification of morphological characteristics of axonal endings and terminal Schwann cell processes in wild-type and *Egfl6* knockout mice.
DOI: https://doi.org/10.7554/eLife.38883.013

**Figure supplement 2.** Ultrastructural examination of lanceolate complexes in wild-type and *Egfl6* knockout mice.
DOI: https://doi.org/10.7554/eLife.38883.014

**Figure supplement 3.** EGFL6 is required for the patterning of lanceolate complexes at the first telogen.
DOI: https://doi.org/10.7554/eLife.38883.015

**Figure supplement 4.** Histological examination of skin tissue morphology in wild-type and *Egfl6* knockout mice.
DOI: https://doi.org/10.7554/eLife.38883.016

**Figure supplement 5.** Skin–nerve preparation and recordings of responses of Aδ-LTMRs.
DOI: https://doi.org/10.7554/eLife.38883.017

## The bulge provides stable epidermal–neuronal interfaces

Unlike most other sensory systems, the hair follicle undergoes dynamic structural changes during its periodic regeneration cycle (*Figure 4—figure supplement 4A and B*). It has remained unclear how hair follicles are able to maintain stable hair follicle–lanceolate complex interactions over the course of this cycle. To gain insight into this question, we first examined the changes that occur in the anatomical structure of the upper-bulge epidermis during the hair regeneration cycle. We found that at the first telogen (P20), the upper-bulge exhibits a single circular peripheral morphology with one hair shaft, while at the first anagen (P35), the upper-bulge perimeter doubles due to the formation of a new hair follicle rostral to the existing hair follicle (old bulge) (*Figure 5A–C*). At the second telogen (P49), the upper-bulge perimeter decreased due to a reduction in the new bulge perimeter. Thus, the external configuration of the rostral side of the upper-bulge epidermis changes dynamically during the hair follicle regeneration cycle, while the caudal aspect remains stable.

We next examined the relationship between the dynamics of hair follicle epidermal morphology and lanceolate complex structure. We measured the signal intensities of immuno-stained lanceolate complex components along the upper-bulge, and assigned average signal intensities to the position angle of hair follicles. At the first telogen, hair follicles exhibited relatively regular signal intensities along the epidermal perimeter, but at the first anagen the new bulge (rostral side) showed areas of low signal intensity, while these remained unchanged at the old (caudal) bulge

**Table 1.** General characteristics of Aδ-LTMRs.

| Outcomes | WT (n = 23 nerves) | *Egfl6* KO (n = 24 nerves) | *P* value |
|---|---|---|---|
| Activation threshold (V) | 3.0 (2.0–4.0) | 3.0 (2.0–6.0) | p=0.778 |
| Conduction velocity (m/s) | 6.6 (5.1–7.7) | 7.7 (5.0–8.6) | p=0.115 |
| Spontaneous activity (imp/s) | 0 (0–0) | 0 (0–0) | p>0.999 |
| Receptive field size (mm$^2$) | 6.5 (4.7–13.2) | 4.8 (2.8–7.3) | p=0.059 |

DOI: https://doi.org/10.7554/eLife.38883.019
The following source data is available for Table 1:
**Source data 1.** Raw numerical data for Table 1.
DOI: https://doi.org/10.7554/eLife.38883.020

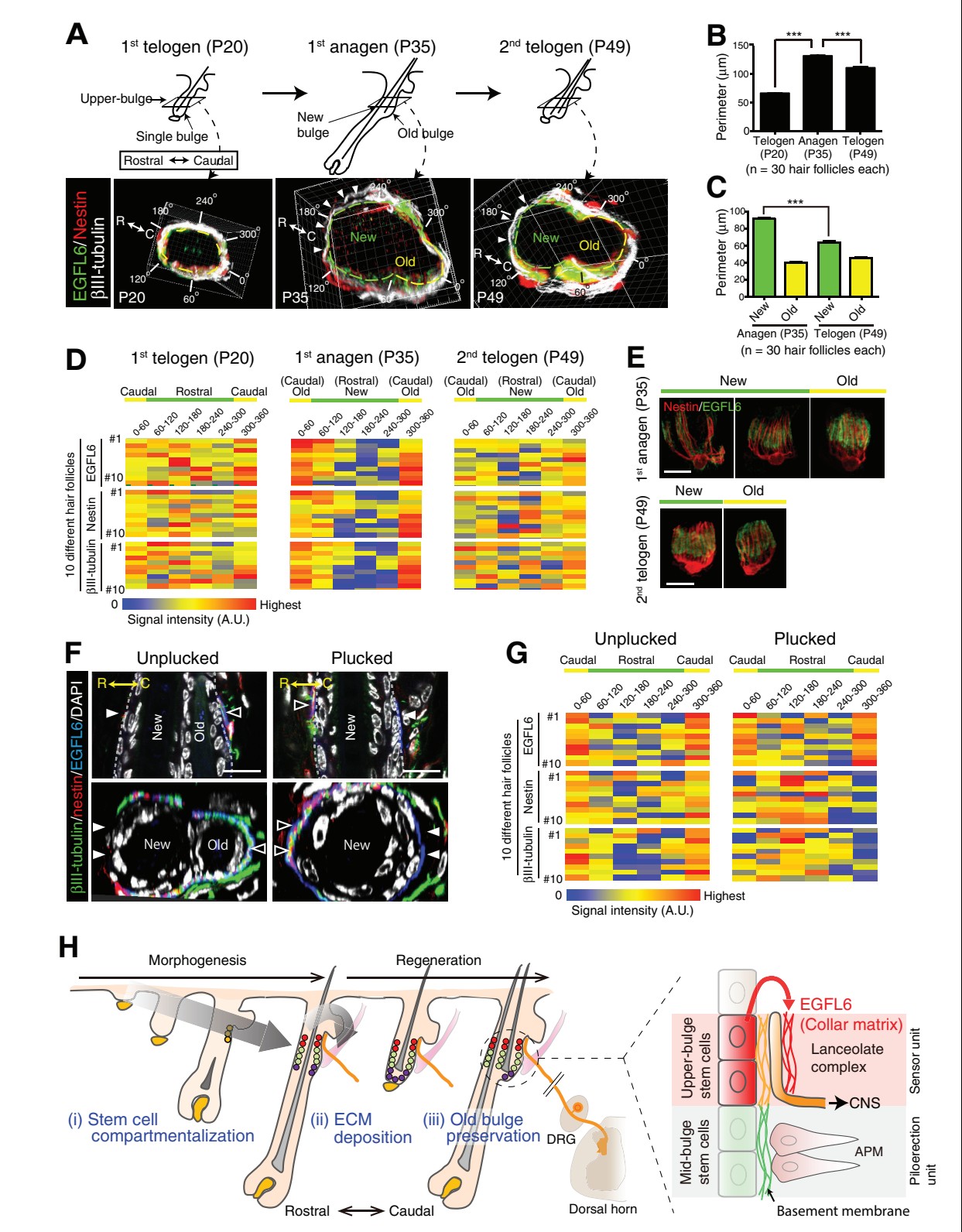

**Figure 5.** The old bulge provides stable epidermal–neuronal interfaces. (A) Structural changes of the upper-bulge during the hair follicle regeneration cycle. Cross-sectional views of the upper-bulge of zigzag hair follicles are shown. Closed arrowheads indicate a large gap of lanceolate complexes. (B) Perimeter length of the upper-bulge and (C) the upper-bulge of new and old bulges. (D) Heat map of the relative signal intensity levels of EGFL6, nestin and βIII-tubulin along the upper-bulge perimeter. Geometry of the upper-bulge perimeter is shown above the heat map. (E) Morphological

*Figure 5 continued on next page*

*Figure 5 continued*

differences of EGFL6/terminal Schwann cell complexes in new and old bulges. (F) Sagittal and transverse sectional views of the upper-bulge in P35 anagen hair follicles with the club hair unplucked and plucked. Open and closed arrowheads indicate areas with and without lanceolate complexes, respectively. (G) Heat map of the relative signal intensity levels of EGFL6, nestin and βIII-tubulin along the upper-bulge perimeter in hair follicles with the club hair unplucked and plucked. (H) Schematic summary of the contribution of epidermal stem cells to sensory unit formation. (i) During development, epidermal stem cells for hair follicle–nerve interactions are induced and compartmentalized in the upper-bulge. (ii) Upper-bulge epidermal stem cells provide specialized ECM and neurogenetic environments for lanceolate complexes. (iii) epidermal stem cells maintain an old bulge at the caudal side of the hair follicle, providing stable hair follicle–nerve interfaces. Scale bars, 10 μm.

DOI: https://doi.org/10.7554/eLife.38883.021

The following source data and figure supplements are available for figure 5:

**Source data 1.** Raw numerical data for *Figure 5* and associated figure supplements.
DOI: https://doi.org/10.7554/eLife.38883.024
**Figure supplement 1.** The old bulge formation provides stable epidermal-neuronal interfaces.
DOI: https://doi.org/10.7554/eLife.38883.022
**Figure supplement 2.** Expression of EGFL family genes in bulge epidermal stem cells of *Egfl6* knockout mice.
DOI: https://doi.org/10.7554/eLife.38883.023

(*Figure 5A and D*). The low-intensity areas at new bulges became narrower in the second telogen. Both Aδ- and C-LTMRs exhibited caudally polarized distribution in the second telogen (*Figure 5—figure supplement 1A*). Consistent with these signal intensity data, EGFL6 and terminal Schwann cells in the old bulge maintained regular parallel morphologies throughout the hair cycle, while those in the new bulge exhibited diverse morphologies and wider widths in the first anagen (*Figure 5E*, *Figure 5—figure supplement 1B*). Thus, the polarized distributions of lanceolate complex components toward the caudal side of the hair follicles are induced as the new bulge structure forms at the rostral side of the hair follicle. The caudal side of the hair follicle, where the old bulge is maintained by quiescent epidermal stem cells, does not change in morphology, suggesting that the old bulge stably preserves lanceolate complexes irrespective of the stage of hair regeneration cycle.

Finally, we plucked a club hair, a retained hair from the previous hair cycle in the old bulge, to test whether the removal of the sensor probe (hair) and subsequent alteration of the old bulge architecture would affect the lanceolate complex structure. The hairs were painted at the first telogen (P20) and the painted hairs (i.e., club hairs) were plucked when the new hair and bulge formed (*Figure 5—figure supplement 1C*). The club hair plucking itself did not compromise the two-bulge epidermal architecture (*Figure 5—figure supplement 1D*). It has been reported that hair plucking induces apoptosis in bulge stem cells within 4.5 hr (*Chen et al., 2015*; *Ito et al., 2002*). Consistent with this, epidermal cells with a bright DAPI signal, an indicator of apoptosis, were observed in the old bulge, but not in the new bulge, within 4 hr after club hair plucking (*Figure 5—figure supplement 1D*). Three days after plucking, the old bulges regressed with a slight decrease in overall follicle perimeter (*Figure 5F*, *Figure 5—figure supplement 1E*), but the new follicles remained, suggesting that the club hair plucking procedure specifically depleted the old bulge epidermal cells through inducing apoptosis and changed the topology of the old bulge. In these plucked hair follicles, axon endings and terminal Schwann cell processes disappeared from the caudal side of the upper-bulge, while ectopic axon endings and terminal Schwann cell processes appeared at the rostral side of the upper-bulge (*Figure 5F and G*). Thus, the maintenance of the old bulge appears to be critical for sustaining caudally polarized lanceolate complex structure.

The observed relocation of lanceolate complexes from the caudal side to the rostral side of the hair follicle can be explained as an active relocation of lanceolate complex processes by old bulge regression, rather a passive sliding of a whole array of lanceolate complex processes toward the rostral side, which could potentially be induced by the simple shrinking of overall follicle perimeter after club hair plucking. This notion is supported by the fact that EGFL6 remained at the caudal side of the plucked hair follicles even without axon endings and terminal Schwann cell processes, while the deposition level of EGFL6 was still low at the rostral side (*Figure 5F and G*). This observation suggests that i) lanceolate complex processes in the old bulge retracted after club hair plucking, but EGFL6 matrix remained in the caudal side, and ii) ectopic lanceolate complex processes were

formed at the rostral side, but EGFL6 deposition required additional time as observed in lanceolate complex morphogenesis.

Together these results indicate that the formation and preservation of the old bulge epidermal structure provides a stable epidermal–neuronal interface and induces an lanceolate complex structure oriented toward the caudal side of the hair follicle. Rutlin et al. (2014) reported that caudally polarized localization of Aδ-LTMRs underlies direction-selective responsiveness of Aδ-LTMRs to hair deflection, suggesting that the unique tissue architecture generated by upper-bulge epidermal stem cells is involved in the directional selectivity of the lanceolate complexes.

## Discussion

Our findings reveal transcriptional and anatomical roles of compartmentalized epidermal stem cells in hair follicle sensory unit formation (*Figure 5H*). Our previous study also showed that the mid-bulge epidermal stem cells express many tendon-related genes and induce the differentiation of APMs and their correct anchorage to the bulge via secretion of the basement membrane protein nephronectin (*Fujiwara et al., 2011*). Deletion of nephronectin moves the muscle near the upper-bulge. The morphology of the lanceolate complex was altered in the nephronectin mutant (*Figure 5—figure supplement 1E–G*), likely due to the changes in mechanical or geometrical environments of the upper-bulge by the mis-location of APMs. Therefore, we propose that the heterogeneous epidermal stem cells are compartmentalized not only for efficient epidermal homeostasis and regeneration (*Schepeler et al., 2014*), but also for defining patterned niches for specific epidermal–dermal interactions, in part through providing different ECM environments.

### Requirement of a unique ECM in touch sensation

We identified a novel electron-dense and amorphous ECM structure, which we named the collar matrix, that ensheathes lanceolate complexes. The deletion of a collar matrix protein, EGFL6, impaired the structure and sensory function of lanceolate complexes. The collar matrix has anatomical and functional features in common with the essential ECM in touch end organs of non-vertebrate organisms, such as the mantle in *C. elegans* (*Emtage et al., 2004*). Thus, electron-dense amorphous ECMs are likely to be the fundamental ECM regulating sensory end organ formation and function in multicellular organisms.

Deletion of EGFL6 did not change the number or length of sensory axonal endings, electrical excitability of Aδ-LTMRs and gross skin histology. These results indicate that defects in tactile responses in the mutants were induced not by the loss or severe damage of touch sensory neurons themselves, but by the changes in the local ECM environment. The ECM can simultaneously function as both a tissue insulator and glue, keeping different cell populations and subcellular structures from intermingling. Indeed, the deletion of EGFL6 modestly but significantly perturbed the parallel stripe pattern of lanceolate endings. Moreover, it has been proposed that the ECM forms stable connections, or tethers, between sensory axon endings and their surroundings to transmit mechanical stimuli to axonal endings (*Zimmerman et al., 2014*). Integrins and integrin signalling genes are required for touch responses of touch receptor neurons in *C. elegans* (*Calixto et al., 2010*). We showed that EGFL6 induces cell adhesion via αv integrins, which are enriched on the collar matrix side of axonal endings of lanceolate complexes in an EGFL6-dependent manner. αv integrins accumulate at synapses of the central nervous system and play key roles in synapse development and function (*Chavis and Westbrook, 2001*; *McCarty et al., 2005*). Thus, EGFL6-αv integrin complexes can potentially be involved in mechanotransduction on the collar matrix side of lanceolate complexes. However, the molecular mechanisms of how EGFL6 regulates lanceolate complex morphology and function are currently unknown. Further genetic analysis, such as an analysis of conditional deletion of αv integrins in sensory nerves or terminal Schwann cells, are needed.

In addition to EGFL6, many other upper-bulge ECM proteins were identified in this study and were located at the upper-bulge with unique tissue distribution patterns. We also showed that lanceolate complexes express multiple integrin receptors that have distinct tissue distribution patterns—accumulated to the collar matrix side and/or the basement membrane side. These results indicate that lanceolate complexes establish topologically complex but well-organized cell–ECM interactions between two molecularly and topologically distinct ECM structures. This rich ECM environment may also cause functional redundancy and compensation among cell–ECM interactions in

the *Egfl6* knockout even without compensatory upregulation of other EGFL family genes (*Figure 5— figure supplement 2*). Further analysis on upper-bulge–specific ECM molecules and their receptors will provide insights into how tactile sensation is regulated by the ECM.

## Compartmentalized epidermal stem cells shape responses to tactile sensation

The biological significance of forming two bulges has not been well explained. A recent report by *Lay et al., 2016* suggests that the two-bulge structure has an advantage for life-long hair follicle epidermal maintenance as it increases the number of epidermal stem cells and maintains stem cell quiescence by holding a keratin 6[+] inner bulge that expresses factors that inhibit stem cell activation. Our study now brings an entirely different perspective on the significance of the old bulge formation. We demonstrated that the formation and preservation of the old bulge provides a stable epidermal-neuronal interface and induces a lanceolate complex structure oriented toward the caudal side of the hair follicle. Caudally polarized expression of BDNF in the bulge epidermis of developing hair follicles regulates caudally enriched localization of Aδ-LTMRs, which underlies direction-selective responsiveness of Aδ-LTMRs (*Rutlin et al., 2014*). Our data show that both Aδ-LTMRs and C-LTMRs become caudally polarized after the formation of the two-bulge structure, indicating that the unique tissue architecture provided by epidermal stem cells is another major determinant of the formation, preservation and function of lanceolate complexes in the hair follicle.

The mechanisms that form and preserve proper innervation in regenerating hair follicle have attracted attention. *Botchkarev et al., 1997* reported the constant number of longitudinal nerve endings in the lanceolate complexes in both natural and hair plucking-induced hair cycle. This invariance of nerve ending number, together with the unique epidermal tissue geometry and dynamics revealed in the present study, may underlie stability and topological reorganization of lanceolate complexes during the hair cycle and in hair plucking–induced old bulge depletion.

In conclusion, our findings give a new perspective on the roles played by heterogeneous and compartmentalized epidermal stem cell populations in more global aspects of organogenesis, beyond their role in epithelial maintenance and regeneration. Our study also provides insights into how sensory organs take advantage of a stable niche provided by a quiescent epidermal stem cell compartment to maintain sensory function within a structurally dynamic tissue environment.

# Materials and methods

**Key resources table**

| Reagent type (species) or resource | Designation | Source or reference | Identifiers | Additional information |
|---|---|---|---|---|
| Genetic reagent (M. musculus) | *Egfl6* KO | MMRRC | 032277-UCD | |
| Genetic reagent (M. musculus) | *Egfl6-H2b-Egfp* BAC transgenic | this paper | | |
| Genetic reagent (M. musculus) | *Ret-eGFP* knock-in | PMID: 17065462 | | Gift from H. Enomoto; *Jain et al., 2006* |
| Genetic reagent (M. musculus) | *Egfl6* KO: *Ret-eGFP* knock-in | this paper | | |
| Genetic reagent (M. musculus) | *Lgr6-GFP-ires-CreERT2* | Jackson Laboratories | 16934 | |
| Genetic reagent (M. musculus) | *Gli1-eGFP* | MMRRC | STOCK:Tg (Gli1-EGFP)DM197 Gsat/Mmucd | |

*Continued on next page*

*Continued*

| Reagent type (species) or resource | Designation | Source or reference | Identifiers | Additional information |
|---|---|---|---|---|
| Genetic reagent (M. musculus) | *Cdh3-eGFP* | MMRRC | STOCK:Tg (Cdh3-EGFP)BK102 Gsat/Mmnc | |
| Cell line (H. sapiens) | Primary keratinocytes | Watt Lab | | Normal foreskin |
| Cell line (H. sapiens) | HEK293 | RIKEN Bioresource Center | RCB1637 | |
| Cell line (H. sapiens) | T98G | JCRB Cell Bank | JCRB9041 | |
| Cell line (H. sapiens) | KG-1-C | JCRB Cell Bank | JCRB0236 | |
| Cell line (H. sapiens) | SF-TY | JCRB Cell Bank | JCRB0075 | |
| Cell line (H. sapiens) | NTI-4 | JCRB Cell Bank | JCRB0220 | |
| Cell line (H. sapiens) | HT-1080 | JCRB Cell Bank | JCRB9113 | |
| Cell line (H. sapiens) | K562 | JCRB Cell Bank | JCRB0019 | |
| Cell line (H. sapiens) | K562-a8 | PMID: 11470831 | | Gift from L. Reichardt; *Chavis and Westbrook, 2001* |
| Cell line (H. sapiens) | 293F | Thermo Fisher | R79007 | |
| Antibody | Rabbit anti-mouse EGFL6 | this paper | CUK-1203–022 | Rabbit polyclonal; against aa 257–550; WB (0.2–2 µg/ml), IHC (1 µg/ml) |
| Antibody | Rabbit anti-mouse nephronectin | this paper | CUK-1192–006 | Rabbit polyclonal; against recombinant full-length mouse nephronectin-His protein; WB (0.4–2 µg/ml), IHC (2 µg/ml) |
| Antibody | Mouse anti-keratin 15 | Santa Cruz Biotechnology | LHK15 (sc-47697) | IHC (1:100) |
| Antibody | Mouse anti-bIII-tubulin | Abcam | 2G10 (ab78078) | IHC (1:250) |
| Antibody | Chicken anti-nestin | Aves Labs | NES0407 | IHC (1:250) |
| Antibody | Rat anti-nestin | Millipore | rat-401 (MAB353) | IHC (1:250) |
| Antibody | Rat anti-mouse CD31 | BioLegend | MEC13.3 | IHC (1:250) |
| Antibody | Chicken anti-GFP | Abcam | ab13970 | IHC (1:2000) |
| Antibody | Rabbit anti-GFP | MBL | 598 | IHC (1:1000) |

*Continued on next page*

*Continued*

| Reagent type (species) or resource | Designation | Source or reference | Identifiers | Additional information |
|---|---|---|---|---|
| Antibody | Rat anti-GFP | Nakarai | GF090R | IHC (1:75) |
| Antibody | Mouse anti-laminin γ1 | Millipore | MAB1914 | IHC (1:500) |
| Antibody | Rat anti-integrin αv | Abcam | RMV-7 | IHC (1:500) |
| Antibody | Hamster anti-mouse integrin αv | BD | H9.2H8 | IHC (1:250) |
| Antibody | Goat anti-human integrin αv | R and D Systems | AF1219 | IHC (1:100) |
| Antibody | Goat anti-mouse integrin α3 | R and D Systems | AF2787 | IHC (1:100) |
| Antibody | Hamster anti-mouse integrin β1 | BioLegend | HMb1-1 | IHC (1:100) |
| Antibody | Goat anti mouse integrin α8 | R and D Systems | AF4076 | IHC (2.5 µg/ml) |
| Antibody | Goat anti-TrkB | R and D Systems | AF1494 | IHC (4 µg/ml) |
| Antibody | Rat anti-CD45-PE-Cy7 | eBioscience | 30-F11 | FACS (1:100) |
| Antibody | Rat anti-TER-119-PE-Cy7 | eBioscience | TER119 | FACS (1:100) |
| Antibody | Rat anti-CD31-PE-Cy7 | eBioscience | 390 | FACS (1:100) |
| Antibody | Rat anti-Sca-1-PerCP-Cy5.5 | eBioscience | D7 | FACS (1:100) |
| Antibody | Rat anti-CD34-eFluor660 | eBioscience | RAM34 | FACS (1:100) |
| Antibody | Rat anti-CD49f-PE | eBioscience | GoH3 | FACS (1:100) |
| Antibody | Mouse anti-integrin α2 | DSHB | P1E6 | Cell adhesion inhibition (10 µg/ml) |
| Antibody | Mouse anti-integrin α3 | DSHB | P1B5 | Cell adhesion inhibition (10 µg/ml) |
| Antibody | Mouse anti-integrin α5 | DSHB | P1D6 | Cell adhesion inhibition (10 µg/ml) |
| Antibody | Rat anti-integrin α6 | BioLegend | GoH3 (313614) | IHC (1:500), Cell adhesion inhibition (10 µg/ml) |

*Continued on next page*

*Continued*

| Reagent type (species) or resource | Designation | Source or reference | Identifiers | Additional information |
|---|---|---|---|---|
| Antibody | Mouse anti-integrin αv | ATCC | L230 | Cell adhesion inhibition (10 µg/ml) |
| Antibody | Mouse anti-integrin β1 | DSHB | P5D2 | Cell adhesion inhibition (10 µg/ml) |
| Recombinant DNA reagent | BAC clone containing mouse *Egfl6* genomic sequence | CHORI BACPAC Resources Center | RP23-124O13 | |
| Sequence-based reagent | qRT-PCR primers | This paper | | See *Table 2* |
| Peptide, recombinant protein | EGF | Peprotech | 100 15 | |
| Peptide, recombinant protein | Insulin | Sigma | I5500 | |
| Peptide, recombinant protein | FLAG-tagged mouse full-length EGFL6 | this paper | | |
| Peptide, recombinant protein | FLAG-tagged mouse full-length EGFL6 (RGE mutant) | this paper | | |
| Peptide, recombinant protein | FLAG-tagged mouse full-length nephronectin | PMID: 21335239 | | *Fujiwara et al., 2011* |
| Peptide, recombinant protein | Human plasma fibronectin | Wako | 063–05591 | |
| Peptide, recombinant protein | Laminin-511-E8 | Nippi | 892011 | |
| Commercial assay or kit | TruSeq Stranded mRNA Sample Prep Kit | Illumina | RS-122–2101 | |
| Commercial assay or kit | HiSeq SR Rapid Cluster Kit v2 | Illumina | GD-402–4002 | |
| Commercial assay or kit | HiSeq Rapid SBS Kit v2 | Illumina | FC-402–4022 | |
| Chemical compound, drug | 0.25% trypsin solution | Nakalai | 35555–54 | |
| Chemical compound, drug | Collagenase type I | Gibco | 17100–017 | |
| Chemical compound, drug | Hydrocortisone | Sigma | H4001 | |
| Chemical compound, drug | Cholera toxin | Sigma | C8052 | |
| Software, algorithm | HiSeq Control Software v2.2.58 | Illumina | | http://www.illumina.com/systems/hiseq_2500_1500/software/hiseq-built-in-software.html |

*Continued on next page*

*Continued*

| Reagent type (species) or resource | Designation | Source or reference | Identifiers | Additional information |
|---|---|---|---|---|
| Software, algorithm | RTA v1.18.64 | Illumina | | http://support.illumina.com/sequencing/sequencing_software/real-time_analysis_rta.html |
| Software, algorithm | bcl2fastq v1.8.4 | Illumina | | http://jp.support.illumina.com/downloads/bcl2fastq_conversion_software_184.html |
| Software, algorithm | FastQC v0.11.3 | Babraham Bioinformatics | | http://www.bioinformatics.babraham.ac.uk/projects/fastqc/ |
| Software, algorithm | FASTX Toolkit v0.0.14 | Hannon Lab | | http://hannonlab.cshl.edu/fastx_toolkit/download.html |
| Software, algorithm | Trim_galore v0.3.3 | Babraham Bioinformatics | | http://www.bioinformatics.babraham.ac.uk/projects/trim_galore/ |
| Software, algorithm | TopHat v2.0.14 | Johns Hopkins University | | https://ccb.jhu.edu/software/tophat/index.shtml |
| Software, algorithm | Cufflinks package v2.2.1 | Trapnell Lab | | http://cole-trapnell-lab.github.io/cufflinks/ |
| Software, algorithm | GSEA software 2.2.2 | Broad Institute | | http://software.broadinstitute.org/gsea/index.jsp |
| Software, algorithm | Bioconductor R with heatmap.2 gplots package | Gregory R. Warnes | | https://www.rdocumentation.org/packages/gplots/versions/3.0.1/topics/heatmap.2 |
| Software, algorithm | Imaris 7.2.1 | Bitplane | | http://www.bitplane.com/imaris/imaris |
| Software, algorithm | Volocity 6.3 | Perkin Elmer | | http://cellularimaging.perkinelmer.com/downloads/ |
| Software, algorithm | Filament Tracer | Bitplane | | http://www.bitplane.com/imaris/filamenttracer |
| Software, algorithm | DAPSYS data acquisition system | DAPSYS | | http://www.dapsys.net/ |
| Other | Piezo-controlled micromanipulator Nanomoter | Kleindiek Nanotechnik | MM3A | |
| Other | Feedback-controlled Peltier device | Intercross, Co. Ltd. | intercross-2000N | |
| Other | RNA-seq Fastq files | this paper | BioProject: PRJNA 342736 | https://www.ncbi.nlm.nih.gov/bioproject/ |

## Mice

*Egfl6* knockout mice (032277-UCD) were obtained from the Mutant Mouse Resource and Research Centers (MMRRC). Lack of *Egfl6* mRNA and EGFL6 protein were confirmed by in situ hybridization

and immunostaining. *Egfl6-H2b-Egfp* BAC transgenic mice were generated as described below. *Ret-eGFP* knock-in mice have been described previously (*Jain et al., 2006*) and kindly provided by H. Enomoto (Kobe University). *Lgr6-GFP-ires-CreERT2* mice were obtained from Jackson Laboratory. *Gli1-eGFP* mice (STOCK Tg(Gli1-EGFP)DM197Gsat/Mmucd) and *Cdh3-eGFP* mice (STOCK Tg (Cdh3-EGFP)BK102Gsat/Mmnc) were obtained from MMRRC. Mouse lines used for transcriptome analysis were backcrossed with C57BL/6N mice more than four times. C57BL/6N and BALB/c mice were obtained from Japan SLC Inc. *Npnt* floxed mice were obtained from Jackson Laboratory and crossed with keratin 5-Cre mice from the Jose Jorcano laboratory. All animal experiments were conducted and performed in accordance with approved Institutional Animal Care and Use Committee protocols (#A2012-03-12).

## Generation of the *Egfl6-H2b-Egfp* mouse line

Egfl6-H2b-Egfp BAC transgenic mice (Accession No. CDB0518T: http://www2.clst.riken.jp/arg/TG% 20mutant%20mice%20list.html) were generated by introducing a BAC carrying a *H2b-Egfp* transgene just before the ATG of the mouse *Egfl6* gene. The BAC clone RP23-124O13, containing the full genomic sequence of *Egfl6*, was obtained from CHORI BACPAC Resources Center. The *H2b-Egfp* fusion gene was introduced just before the first coding ATG of the *Egfl6* gene in the BAC clone using a two-step selection BAC recombineering protocol (*Warming et al., 2005*). The modified BAC construct was purified with a NucleoBond Xtra Midi kit (Macherey-Nagel) and injected into the pro-nuclei of fertilized one-cell mouse eggs derived from the breeding of BDF1 and C57BL/6N mice. Potential founder mice were screened by genomic PCR of ear biopsy DNA with following primers: 5'- CCAAGGTCCTGACCAGCGAAG-3' and 5'- CCTTAGTCACCGCCTTCTTGGAG −3' (product size: 163 base pairs). These primers were also used for routine genotyping of this mouse line. The transgene-positive founder mice were bred with wild-type C57BL/6N mice and some of the offspring were used for immunostaining of eGFP to confirm the expression and nuclear localization of the H2B-eGFP protein. We have established 16 nuclear-GFP-positive transgenic mouse lines and confirmed the GFP expression in the upper-bulge in 13 mouse lines.

## Cell lines

Cell lines used in this study were 293F cells (Thermo Fisher), human primary keratinocytes (Watt lab), human embryonic kidney epithelial cell line HEK293 (RIKEN Bioresource Center), human glioblastoma cell line T98G (JCRB Cell Bank), Human glioma cell line KG-1-C (JCRB Cell Bank), human skin fibroblast cell line SF-TY (JCRB Cell Bank), human embryonic fibroblast cell line NTI-4 (JCRB Cell Bank), human sarcoma cell line HT-1080 (JCRB Cell Bank), human myelogenous leukemia cell line K562 (JCRB Cell Bank) and human myelogenous leukemia cell line transfected with human integrin α8 cDNA K562-a8 (Gift from L. Reichardt).

Human primary keratinocytes were cultured on irradiated J2 feeder cells with DMEM/Ham's F12 medium supplemented with 10% FBS, 0.5 μg/ml hydrocortisone (Sigma), 0.1 nM cholera toxin (Sigma), 10 μg/ml EGF (Peprotech), 2 mM GlutaMax (Invitrogen) and 5 μg/ml Insulin (Sigma) at 37°C under 5% $CO_2$. Other cell lines were cultured according to the cell culture instructions from each source cell bank or company.

Cell lines were purchased directly from the JCRB Cell Bank, which regularly performs cell line authentication, and then immediately used for this study. Human HEK293 and 293F were obtained directly from RIKEN Bioresource Center and Thermo Fisher, respectively, and then immediately used for this study. Human K562-a8 were constantly selected with G418 for their transgene expression. Cell lines from JCRB, RIKEN Bioresource Center and Thermo Fisher were negative for Mycoplasma contamination.

## Antibodies

Primary antibodies used in this study and their dilution ratios are listed in Key Resources Table.

## Generation of antibodies

Rabbit antiserum to mouse EGFL6 was generated by immunizing rabbits with a Myc-His-tagged Mucin-MAM domain of recombinant mouse EGFL6 (S257-G550). The antigen was expressed in the 293F mammalian expression system (Invitrogen) and purified from the supernatant of the transfected

cells with a Ni-NTA column. The antibody in the antiserum was affinity-purified with an antigen conjugated CNBr-activated Sepharose 4B. The specificity of the antibody to mouse EGFL6 was confirmed by the absence of the antibody immunoreactivity to tissue samples from *Egfl6* knockout mice. Rabbit antiserum to mouse nephronectin was generated by the same method described above. Recombinant flag-tagged full-length mouse nephronectin was expressed in 293F cells and purified with a FLAG-M2 immuno-affinity column (Sigma). The antibody in the antiserum was affinity-purified with a CNBr-activated Sepharose 4B conjugated with purified His-tagged mouse full-length nephronectin. The specificity of the antibody to mouse nephronectin was confirmed by the absence of the antibody immunoreactivity to tissue samples from *Npnt* knockout mice.

## FACS

Mouse adult dorsal telogen keratinocytes were isolated and stained for cell surface markers as described previously (*Fujiwara et al., 2011*) with some modifications. We utilized *Lgr6-GFP-ires-CreERT2*, *Gli1-eGFP* and *Cdh3-eGFP* mice to fluorescently visualize the epidermal stem cells in the lower-isthmus (*Lgr6*$^+$), upper-bulge (*Gli1*$^+$), and hair germ (*Cdh3*$^+$) with eGFP. Dissected dorsal skin of the 8-week-old female mice was treated with 0.25% trypsin solution (Nakalai tesque) at 37°C for 1 hr. The epidermal tissue was scraped off from the dermal tissue with a scalpel. This epidermis separation protocol leaves hair germ cells in the dermal tissue. Effectiveness of this tissue separation was verified by the following qRT-PCR, RNA-seq and immunohistochemical analyses with compartment-specific markers as described below. For the sorting of lower-isthmus (*Lgr6*-eGFP$^+$), upper-bulge (*Gli1*-eGFP$^+$) and mid-bulge (CD34$^+$) epidermal stem cells, the separated epidermis was minced with scalpels and mixed with repeated pipetting to make a single cell suspension. To deplete haematopoietic and endothelial cells (lineage-positive cells; Lin$^+$), the cell suspension was stained with PE-Cy7-conjugated antibodies for CD45, TER-119 and CD31. To sort the target cells, the cell suspension was also stained with Sca-1-PerCP-Cy5.5, CD34-eFluor660, CD49f (integrin α6)-PE. Sca-1 was used to remove epidermal basal cells of the interfollucular epidermis and follicle infundibulum (*Jensen et al., 2008*). Cells were sorted with a FACSAria II according to the expression of cell surface markers, after gating out dead and Lin$^+$ cells (*Figure 1—figure supplement 1A*).

Hair germ epidermal cells (*Cdh3*-eGFP$^+$) were sorted from the remaining dermal tissue since the most hair germ epidermal cells were retained in the dermal tissue after the separation of the epidermis. The dermal tissue was minced with scalpels and incubated with 2 mg/ml of collagenase type I at 37°C for 2 hr with gentle mixing. Single cell suspension was obtained by repeated pipetting. The cell suspension was stained with the same antibodies used above and subjected to the sorting procedure.

The purity of sorted cells was confirmed by qRT-PCR with compartment-specific genes: *Lrig1* (junctional zone and lower-isthmus), *Lgr6* (lower-isthmus and upper-bulge), *Gli1* (upper-bulge and hair germ), *Bdnf* (upper-bulge), *Cd34* (mid-bulge), *Cdh3* (hair germ) (*Figure 1—figure supplement 1B*). While the expression of *Gli1*-eGFP was detected in both upper-bulge and hair germ regions (*Figure 1B*), our cell isolation and sorting methods gave highly pure upper-bulge cells and hair germ cells, respectively. For example, a marker for upper-bulge cells, *Bdnf*, was highly enriched in *Gli1* + upper bulge cells, but not in *Cdh3* +hair germ cells (*Figure 1—figure supplement 1B*). On the other hand, the expression of *Cdh3*, a marker for hair germ cells, was high in the *Cdh3* + hair germ population, but was very low in the *Gli1* +upper bulge population (*Figure 1—figure supplement 1B*). We identified *Spon1* as a gene exclusively expressed in the hair germ from our gene expression profiling of isolated stem cell populations and confirmed that it was not detected in the *Gli1*+ upper bulge population (*Figure 1—figure supplement 1C*). Subsequent protein tissue-localization analysis of compartment-specific genes, which were identified in our gene expression profiling, further confirmed the validity and purity of sorted cells. For example, gene products of upper-bulge-enriched ECM genes, including *Aspn*, *Crispld1*, *Egfl6*, *Col4a3* and *Col4a4*, were deposited in the upper-bulge (*Figure 2A*), while SPON1 protein was localized in the hair germ region, but not in the upper-bulge (Figure 1—figure supplement 1C). Collectively, our data demonstrate that the purity of each isolated stem cell population was very high.

## qRT-PCR

Total RNA was extracted from the sorted cells using an RNeasy micro kit (Qiagen). qRT-PCR was performed to determine target cell populations using Power SYBR Green PCR Master Mix (Life Technologies) with specific primer sets (*Table 2*) on 7900HT real-time PCR system (Applied Biosystems) or CFX384 (Bio-Rad).

## RNA library preparation and sequencing

For each library, isolated total RNA samples were quantified with the Qubit RNA HS Assay Kit on a Qubit Fluorometer (Thermo Fisher Scientific), and their qualities were analyzed with the RNA 6000 Pico Kit on a 2100 Bioanalyzer (Agilent Technologies). Qubit measurements were used to ensure that each library was prepared with 10 ng of total RNA. Library preparation was processed following the TruSeq Stranded mRNA Sample Prep Kit (Illumina) protocol till adapter ligation, except that the duration of the initial RNA fragmentation was shortened to 7 min. The adapter-ligated cDNA was amplified with 14 PCR cycles. The prepared libraries were sequenced using the Rapid Run mode with 80 cycles on the HiSeq1500 (Illumina), operated by HiSeq Control Software v2.2.58 using HiSeq SR Rapid Cluster Kit v2 (Illumina) and HiSeq Rapid SBS Kit v2 (Illumina). Base calling was performed with RTA v1.18.64 and the fastq files were generated with bcl2fastq v1.8.4 (Illumina).

## Mapping and expression quantification

Qualities of the RNA-seq reads were evaluated with FastQC v0.11.3. The program fastq_quality_filter, part of the FASTX Toolkit v0.0.14, was used to trim bases with a quality value below 30. Additionally, if more than 20% of the bases of a read were removed by this trimming procedure, then the whole read was discarded. Once this quality trim was completed, Trim_galore v0.3.3 was executed to remove adapter sequences from the 3'-ends of reads, and to discard reads that were shorter than 50nt. The processed reads were mapped to the mm10 mouse genome assembly, obtained from iGenomes (http://support.illumina.com/sequencing/sequencing_software/igenome.html), using the splice-aware aligner TopHat v2.0.14 with default parameter settings. Reads mapping to ribosomal DNA accounted for ~1% of the total number of reads in each library, and were removed. Details for each library concerning sequencing and read statistics, including the total amount that uniquely mapped to the genome, are displayed in *Table 3*.

Gene expression quantification was performed using the Cuffdiff program in the Cufflinks package v2.2.1, with the frag-bias-correction and multi-read-correction options enabled. Cuffdiff used the mm10 gene model obtained from iGenomes to estimate the number of fragments that originated from individual genes. For each gene, in addition to this produced raw count data, Cuffdiff also calculated normalized expression values, which are referred to as fragments per kilobase of transcript per million mapped reads (FPKM).

## GO and clustering analysis

FPKM values from RNA-seq data were analysed by Gene Ontology (GO) term enrichment analysis. Genes with low expression (FPKM <5) were filtered out and considered not expressed. Then the remaining FPKM values were log2-transformed for further analysis. To characterize global gene expression profiles of each epidermal stem cell population, gene set enrichment analysis (GSEA) was performed using GSEA software (2.2.2 from Broad Institute) with GO biological processes as a gene set collection (c5.bp.v5.1.symbols.gmt). In this analysis, pairwise differences between basal epidermal stem cells and each epidermal stem cell population were examined. The normalized enrichment score (NES) and False Discovery Rate (FDR) were calculated for each gene set and compared among cell populations using a bioconductor R with heatmap.2 gplots package.

## Immunohistochemistry and imaging

Whole-mount immunostaining and horizontal imaging of mouse dorsal skin were performed as described previously (*Fujiwara et al., 2011*). Vertical whole-mount imaging of dorsal and whisker skin was performed as described below. Mouse skin tissues were dissected and fixed with 4% paraformaldehyde/PBS for 1 hr at 4°C. Fixed tissues were washed and embedded in OCT compound and frozen on liquid nitrogen. Vertical skin sections (150 μm thick) were made using a cryostat (Leica) and washed with PBS to remove remnant OCT compound. Skin sections were blocked with a

**Table 3.** RNA-seq read counts and mapping statistics.

| | Raw Read Count | Trimmed Read Count | rDNA Read Count | Multi-Mapped Read Count | Uniquely mapped Read Count |
|---|---|---|---|---|---|
| Mouse_Basal_Cells_Rep1 | 11,757,494 | 11,278,061 | 148,700 | 170,077 | 10,456,332 |
| Mouse_Basal_Cells_Rep2 | 11,245,045 | 10,668,752 | 73,729 | 134,777 | 10,051,376 |
| Mouse_Basal_Cells_Rep3 | 11,313,239 | 10,747,103 | 108,596 | 146,602 | 10,089,518 |
| Mouse_Bulge_StemCells_Rep1 | 11,735,034 | 11,181,377 | 64,614 | 183,557 | 10,520,864 |
| Mouse_Bulge_StemCells_Rep2 | 11,917,049 | 11,515,851 | 54,154 | 175,176 | 10,889,932 |
| Mouse_Bulge_StemCells_Rep3 | 13,417,100 | 12,895,113 | 88,655 | 218,495 | 12,090,872 |
| Mouse_Cdh3+_Cells_Rep1 | 10,875,593 | 10,409,342 | 68,193 | 177,407 | 9,574,901 |
| Mouse_Cdh3+_Cells_Rep2 | 11,565,616 | 10,993,093 | 65,001 | 185,981 | 10,079,620 |
| Mouse_Cdh3+_Cells_Rep3 | 11,405,265 | 10,746,768 | 76,143 | 206,384 | 9,790,096 |
| Mouse_Cdh3+_Cells_Rep4 | 11,759,881 | 11,186,543 | 85,902 | 191,154 | 10,274,371 |
| Mouse_Gli1+_Cells_Rep1 | 12,063,488 | 11,478,537 | 57,630 | 178,646 | 10,751,521 |
| Mouse_Gli1+_Cells_Rep2 | 19,762,234 | 18,804,713 | 178,685 | 303,848 | 17,446,776 |
| Mouse_Gli1+_Cells_Rep3 | 19,543,117 | 18,568,695 | 122,017 | 287,214 | 17,378,904 |
| Mouse_Lgr6+_Cells_Rep1 | 20,044,455 | 19,058,807 | 121,507 | 307,804 | 17,798,594 |
| Mouse_Lgr6+_Cells_Rep2 | 19,756,354 | 18,736,071 | 197,458 | 298,652 | 17,386,391 |
| Mouse_Lgr6+_Cells_Rep3 | 19,639,299 | 18,680,969 | 217,502 | 298,146 | 17,210,635 |

DOI: https://doi.org/10.7554/eLife.38883.026

blocking buffer (0.5% skim milk/0.25% fish skin gelatin/0.5% Triton X-100/PBS) for 1 hr at 4°C and then incubated with primary antibodies diluted in blocking buffer overnight at 4°C. Skin samples were washed with 0.2% Tween 20/PBS for 4 hr and then incubated with DAPI and secondary antibodies diluted in blocking buffer overnight at 4°C. Finally, skin samples were washed with 0.2% Tween20/PBS for 4 hr at 4°C and mounted with BABB clearing solution. Images were acquired using a Leica TSC SP8. Z stack maximum projection images and three dimensional reconstructed images of skin whole-mount preparation were produced using Imaris 4D rendering software (Bitplane) and Volocity 3D imaging software (Perkin Elmer).

## Transmission electron microscopy

Dissected skin tissues were immediately immersed in 2% fresh formaldehyde and 2.5% glutaraldehyde in 0.1M sodium cacodylate buffer (pH 7.4), sliced into 0.5–1 mm-thin sections with a scalpel and fixed for 2 hr at room temperature. After washing three times with 0.1M cacodylate buffer (pH 7.4) for 5 min, tissues were post-fixed with ice-cold 1% $OsO_4$ in the same buffer for 2 hr. Samples were rinsed with distilled water, stained with 0.5% aqueous uranyl acetate for 2 hr or overnight at room temperature, then dehydrated with ethanol and propylene oxide, and embedded in Poly/Bed 812 (Polyscience). Ultra-thin sections were cut, doubly-stained with uranyl acetate and Reynold's lead citrate, and viewed with a JEM 1010 or JEM 1400 plus transmission electron microscope (JEOL) at an accelerating voltage of 100 kV.

## Immunoelectron microscopy

Skin samples were dissected, fixed and stained according to the whole-mount immunostaining method. In-house rabbit EGFL6 antibody was used as the primary antibody and Alexa Fluor 488 FluoroNanogold-anti rabbit IgG (Nanoprobes) (1:200 dilution) was used for the secondary antibody. After confirming the localization of Alexa Fluor 488 labelled EGFL6 under a fluorescent microscope, the samples were fixed for 1 hr in 1% glutaraldehyde in PBS. GoldEnhance EM (Nanoprobes) was used in accordance with the manufacturer's protocol to enlarge the size of gold particles. Samples were then embedded in resin (poly/bed 812; Polyscience). Ultrathin sections were stained with uranyl acetate and lead citrate before observation with an electron microscope (JEM 1400 plus; JEOL).

## Quantification of axon and terminal Schwann cell processes

Neural filaments and terminal Schwann cell protrusions were visualized, measured and analysed using the Imaris software program FilamentTracer (Bitplane). To detect Aδ- and C-LTMRs and terminal Schwann cells in *Egfl6* knockout mice, we crossed *Egfl6* knockout mice with *Ret*-eGFP knock-in mice (*Jain et al., 2006*) that express eGFP in C-LTMRs (*Li et al., 2011*). Skin whole-mounts were immunostained for TrkB, eGFP and nestin to visualize Aδ- and C-LTMRs and terminal Schwann cells, respectively. To examine the structure of lanceolate complexes, we analysed old bulge regions, but not new bulge regions, since the new bulge regions tend to show a small amount of EGFL6 and an irregular shape of lanceolate complexes. Axon endings and terminal Schwann cell processes were automatically detected in three dimensionally reconstructed immunohistochemical images using FilamentTracer. Traced axonal endings and terminal Schwann cell protrusions were visualized as fixed-diameter cylinders in green (*Ret*-eGFP), red (TrkB) and white (nestin). Length, number and width of axon endings and terminal Schwann cell processes were automatically measured by the software. Overlapping filament points were detected three-dimensionally using the cylinder display mode. For the analysis of axon and terminal Schwann cell processes in first telogen samples, we used wild-type and *Egfl6* knockout mice. All LTMRs and Aδ-LTMRs were visualized with βIII-tubulin antibody and TrkB antibody, respectively. Since TrkB⁻ axonal endings in zigzag hair follicles are C-LTMRs (*Li et al., 2011*), we assigned them as C-LTMRs in this analysis.

For the measurement of the upper-bulge perimeter, immunostained whole-mount z-stack images were reconstituted to 3D images with Imaris 7.2.1 software. An Imaris Oblique Slicer function was used to make a plane cutting through the region of terminal Schwann cell processes and this plane was defined as a measurement layer. The perimeter of the hair follicle was traced, and the length of the trace was automatically measured by the software.

## Quantification of the expressions of *Egfl6*-H2BeGFP and αv integrins

GFP expression in *Egfl6*-H2BeGFP reporter mice was used to examine the expression of *Egfl6* transcripts. Adult telogen dorsal skin of *Egfl6*-H2BeGFP mice were immunostained for GFP, keratin 14 (marker for basal keratinocytes) and nestin (marker for terminal Schwann cells), with DAPI nuclear counter staining, as described above. Vertical 3D images of zigzag hair follicles were obtained. The number of total and GFP-positive basal epidermal cells, terminal Schwann cells and other dermal cells at the upper-bulge were counted with reference to GFP, keratin 14, nestin and DAPI staining using Imaris software. Twenty-one zigzag hair follicles from three mice were used. The total number of cells counted were 1010 basal epidermal stem cells, 91 terminal Schwann cells and 319 other dermal cells.

To investigate the expression of *Egfl6* in sensory nerves of lanceolate complexes, we examined the GFP expression in the adult thoracic dorsal root ganglia (DRGs) of *Egfl6*-H2BeGFP mice since the nuclei of skin sensory nerves are located in the DRGs. DRGs were isolated and fixed with 4% PFA for 1 hr at 4°C. Whole DRG tissues were immunostained for βIII-tubulin (marker for DRG neurons) and GFP, with DAPI counter stain. Single sectional plane images were obtained as described above. Twenty-five DRGs from three mice were examined and 2,138 DRG neurons were quantified for their nuclear GFP expression.

To quantify the expression level of αv integrins, we stained telogen adult dorsal skin of wild-type and *Egfl6* knockout mice for EGFL6, αv integrin and nestin, with DAPI counter stain. Signal intensity of αv integrins at the outer surface of the lanceolate complexes was measured using Fiji ImageJ 1.0.

Thirteen hair follicles from three wild-type mice and 16 hair follicles from three *Egfl6* knockout mice were used for quantification.

## Ex vivo skin–nerve preparations

Receptive properties of cutaneous Aδ-LTMRs were studied using mouse hindlimb skin–saphenous nerve preparations ex vivo (*Zimmermann et al., 2009*). Ten- to twelve-week-old male mice, which were in the resting hair growth phase, were euthanized by inhalation of $CO_2$ gas and the preparations were quickly isolated from the left hindlimb. The excised skin was oriented with the dermis side up in the test chamber and affixed with pins. The preparation was maintained at $32 \pm 0.3°C$ (pH 7.4) during the experiment under superfusion with modified Krebs-Henseleit solution (Krebs solution), which contained 110.9 mM NaCl, 4.7 mM KCl, 2.5 mM $CaCl_2$, 1.2 mM $MgSO_4$, 1.2 mM $KH_2PO_4$, 25 mM $NaHCO_3$ and 20 mM glucose. The perfusate was continuously bubbled and equilibrated with a gas mixture of 95% $O_2$ and 5% $CO_2$. The hindlimb skin of mouse is thinner than dorsal skin.

## Recordings of Aδ-LTMRs

Single Aδ-LTMR was searched and identified when it fulfilled the following criteria: 1) fibers with conduction velocity between 2–10 m/s, 2) fibers responding well to innocuous touch stimulus applied by a blunt glass rod compared to noxious vertical indentation of the skin (compression), 3) fibers without mechanical stimulus intensity-dependent increase in the firing rate, 4) fibers exhibiting a brisk, rapidly adapting discharge at the onset of a supramaximal constant force stimulus (*Koltzenburg et al., 1997*), 5) fibers not responding to noxious heat and cold stimuli (*Figure 4—figure supplement 5*) (*Zimmermann et al., 2009*), and 6) fibers showing a stronger excitation typically in the ramp phase of mechanical stimulation when the stimulus probe was horizontally moving compared to the hold phase, as shown in *Figure 4—figure supplement 5*. The fibers identified by these criteria in wild type dermis-up skin-nerve preparation exhibited von Frey hair (vFH) threshold values of ≤0.51–3.23 mN (median: 1.3 mN, IQR: 0.8–2.0 mN) (*Figure 4D*), which fell into the range of previously reported vFH threshold values of Aδ-LTMRs (or D-hair) with the same dermis-up method (<1–5.7 mN), but not that of high-threshold Aδ-mechano nociceptors (5.7–128 mN) (*Zimmermann et al., 2009*).

When an Aδ-LTMR was identified, the receptive field (RF) was stimulated with electronic pulses via a bipolar stimulating electrode (frequency of 0.5 Hz, pulse duration of 100 µs and stimulus intensity of <50 V) to measure the conduction velocity of a fiber. The conduction velocity was calculated from the distance and conduction latency of a spike induced by electrical stimulation of the RF. Spontaneous activity of Aδ-LTMRs was analysed for 20 s just before a series of touch mechanical stimulation. Distribution (size and location) of the RFs was mapped on a standardized chart. The size of the RF was measured by calculating the number of pixels in the RF that was drawn on a chart with Image J software. All of the data were stored in a computer via an A/D converter (Power Lab/16 s, ADInstruments) with a sampling frequency of 20 kHz. Action potentials were analysed on a computer with the DAPSYS data acquisition system (http://www.dapsys.net). Quantitative mechanical, cold and heat stimuli were then applied to the identified RF in the following order: 1) ramp-and-hold touch mechanical stimulation, 2) cooled from 32°C to 8°C and 3) heated from 32°C to 50°C.

## Mechanical stimulation

At first, the mechanical sensitivity of Aδ-LTMRs was semi-quantitatively analysed using a series of self-made von Frey hairs (vFHs: 0.5–17.6 mN, 0.5 mm in diameter). The strength of the weakest filament that caused a mechanical response was taken as the threshold.

For quantitative analysis of the mechanical sensitivity of a fiber, a light touch stimulus was applied using a piezo-controlled micromanipulator (Nanomoter MM3A, Kleindiek Nanotechnik, Reutlingen, Germany). The stimulator had a glass probe with a spherical tip (diameter: 0.5 mm). After placing the probe tip to the most sensitive point on the identified RF, a series of ramp-and-hold touch mechanical stimuli were applied by driving the stimulator with a pre-programed computerized protocol. The tip was moved alternately in rostral and caudal directions with displacement in a progressively increasing manner (4–1280 µm) at a speed of 300 µm/s (*Figure 4—figure supplement 5*). The holding time was set for 2 s between the rostral and caudal movement of the probe tip.

Since all the Aδ-LTMRs were spontaneously silent at the beginning of the experiment without any intentional stimuli, the mechanical response threshold was defined as the displacement that induced the first discharge during the stimulation protocol. If a fiber showed no action potentials during the course of the protocol, even though it exhibited firing as a result of manual touch with a blunt glass rod, then the mechanical threshold was defined to be 1280 µm. The magnitude of the touch response was represented by the number of spikes evoked during the rostral or caudal movement of the probe (i.e. ramp phases), but not during holding phases of 2 s without probe movement.

## Cold and heat stimulation

We applied ramp-shaped thermal stimuli to the RF using a feedback-controlled Peltier device (inter-cross-2000N, Intercross, Co. Ltd., Japan) with a small probe (diameter: 1 mm). From a baseline temperature of 32°C, the RF was gradually cooled down to 8°C over 40 s or heated up to 50°C over 30 s at a constant rate of 0.6 °C/s. Original temperature traces are shown in *Figure 4—figure supplement 5B*.

## Preparation of extracellular matrix proteins

FLAG-tagged mouse full-length EGFL6 was purified as described below. 293F cells were transfected with the *Egfl6-flag* expression vector and cultured for 4 days according to the manufacturer's instruction (Invitrogen). The supernatant was collected and incubated with anti-FLAG M2 affinity gel (Sigma) overnight at 4°C. The FLAG affinity gel was collected in an empty Econo-Column (Bio-Rad) and washed with PBS. Bound protein was eluted with 100 µg/ml FLAG peptide in PBS. The eluted protein solution was dialyzed with PBS. Protein concentration was measured with a Pierce BCA Protein Assay Kit using bovine serum albumin (BSA) as a control. The expression vector of RGE-mutant *Egfl6* was generated using the Toyobo KOD-Plus-Mutagenesis kit (Toyobo). FLAG-tagged mouse full-length nephronectin was purified as described previously (*Fujiwara et al., 2011*). Human plasma fibronectin (Wako) and laminin-511-E8 (Nippi) were purchased from the companies indicated.

## Solid-phase cell adhesion assays

Solid-phase cell adhesion assays were performed as described previously (*Fujiwara et al., 2011*) with minor modifications. Briefly, 96-well cell culture plates were coated with purified ECM proteins and blocked with 1% heat-denatured BSA. Human primary and cultured cell lines were suspended in serum-free DMEM, plated on the coated plates, and incubated for 30 min in a $CO_2$ incubator at 37°C. Attached cells were fixed and stained with 0.5% crystal violet in 20% methanol for 15 min. The cell-bound crystal violet was extracted with 1% SDS solution and the absorbance was measured at 595 nm. To inhibit cell adhesion activity of different integrins in cell adhesion assays, integrin function blocking antibodies were added to cell suspension before seeding to the coated wells. The cell suspension with the antibodies was incubated for 20 min and then plated on the coated dishes. The function-blocking integrin antibodies used in this study were listed below: integrin α2 (P1E6), α3 (P1B5), α5 (P1D6), α6 (GoH3), αv (L230), and β1 (P5D2). P1E6, P1B5, P1D6, and P5D2 were purchased from the Developmental Studies Hybridoma Bank of the University of Iowa. GoH3 and L230 were purchased from BioLegend and ATCC, respectively. Cell lines used in the assays were human primary keratinocytes, HEK293, T98G, KG-1-C, SF-TY, NTI-4, HT-1080, K562 and K562-a8.

## Hair plucking assays

Dorsal hairs of 20-day-old anesthetised BALB/c mice were dyed using a neon orange hair colour cream for 30 min. Mice were maintained till the tip of new growing undyed hairs could be observed at the skin surface under a dissection microscope (P32-35). The dyed long club hairs, but not short emerging undyed new hairs, were plucked using tweezers under anaesthesia in an area of 10 × 15 mm. Three days after club hair plucking (P36-38), dorsal skin tissue samples containing both plucked and unplucked hair follicles were collected and subjected to whole-mount immunostaining. Plucked and unplucked skin areas could be easily distinguished by hair colour: the unplucked area showed dyed orange hair colour and the plucked area showed undyed white hair colour. In skin tissue samples, hair shafts could be identified by their autofluorescence. This allowed us to distinguish plucked and unplucked hair follicles within tissue samples under a confocal microscope. Zigzag hair follicles were selected for the analysis. Hair follicle types were distinguished by the following criteria under

light-field illumination of 150 µm-thick tissue sections. Zigzag hair follicles are the most abundant follicle type in adult mouse dorsal skin (~81%) (*Driskell et al., 2009*). Growing new zigzag hairs in plucked and unplucked hair follicles have one row of medulla cells and the smallest diameter (9.58 ± 0.90 µm, n = 32 hair follicles) at the bulge region in comparison to other hair types. Awl and auchene hair follicles together make up ~17% of the dorsal skin hair follicles. Growing awl/auchene hairs in plucked and unplucked hair follicles have more than three rows of medulla cells and have a large hair shaft diameter (15.7 ± 1.95 µm, n = 27 hair follicles) at the bulge region. Guard hair follicles are rare (~1% of the adult dorsal skin hair follicles) and can be clearly identified by their prominent large follicle diameter.

## Statistical analysis

Statistical parameters including the numbers of samples and replicates, types of statistical analysis and statistical significance are indicated in the Results, Figures and Figure Legends. p values: *$p<0.05$; **$p<0.01$; ***$p<0.001$.

## Acknowledgments

We thank Shigehiro Kuraku, Yuichiro Hara, Osamu Nishimura of the Laboratory for Phyloinformatics, RIKEN for help in RNA-seq and bioinformatics; Hideki Enomoto for Ret-GFP mice; RIKEN Kobe light microscopy and animal facilities for technical assistance; Shigeo Hayashi and Douglas Sipp for their critical reading of the manuscript. We also thank the members of the Fujiwara laboratory for valuable reagents and discussion. This work was funded by RIKEN intramural funding, JSPS KAKENHI (25122720), Uehara Memorial Foundation, Takeda Science Foundation and Cosmetology Research Foundation (all to HF). C-CC was a recipient of the RIKEN-NSC Taiwan Fellowship. FMW acknowledges financial support from the Medical Research Council, BBSRC and Wellcome Trust.

## Additional information

### Funding

| Funder | Grant reference number | Author |
|---|---|---|
| RIKEN | Intramural grant | Hironobu Fujiwara |
| Japan Society for the Promotion of Science | 25122720 | Hironobu Fujiwara |
| Uehara Memorial Foundation | | Hironobu Fujiwara |
| Takeda Science Foundation | | Hironobu Fujiwara |
| Cosmetology Research Foundation | | Hironobu Fujiwara |
| Medical Research Council | | Fiona M Watt |
| Biotechnology and Biological Sciences Research Council | | Fiona M Watt |
| Wellcome | | Fiona M Watt |

The funders had no role in study design, data collection and interpretation, or the decision to submit the work for publication.

### Author contributions

Chun-Chun Cheng, Conceptualization, Data curation, Formal analysis, Validation, Investigation, Visualization, Methodology, Writing—original draft, Writing—review and editing; Ko Tsutsui, Data curation, Formal analysis, Validation, Investigation, Visualization, Methodology, Writing—review and editing; Toru Taguchi, Data curation, Formal analysis, Validation, Investigation, Visualization, Methodology, Writing—original draft, Writing—review and editing; Noriko Sanzen, Formal analysis, Investigation, Visualization; Asako Nakagawa, Kisa Kakiguchi, Chiharu Tanegashima, Investigation; Shigenobu Yonemura, Investigation, Methodology; Sean D Keeley, Data curation, Formal analysis;

Hiroshi Kiyonari, Yasuhide Furuta, Resources, Investigation; Yasuko Tomono, Resources; Fiona M Watt, Resources, Funding acquisition; Hironobu Fujiwara, Conceptualization, Resources, Data curation, Formal analysis, Supervision, Funding acquisition, Validation, Investigation, Visualization, Methodology, Writing—original draft, Project administration, Writing—review and editing

### Author ORCIDs
Fiona M Watt [iD] http://orcid.org/0000-0001-9151-5154
Hironobu Fujiwara [iD] http://orcid.org/0000-0003-0883-3384

### Ethics
Animal experimentation: All animal experiments were conducted and performed in accordance with approved Institutional Animal Care and Use Committee protocols (#A2012-03-12).

### Decision letter and Author response
Decision letter https://doi.org/10.7554/eLife.38883.031
Author response https://doi.org/10.7554/eLife.38883.032

## Additional files

### Supplementary files
• Transparent reporting form
DOI: https://doi.org/10.7554/eLife.38883.027

### Data availability
Fastq files of RNA-seq data have been submitted to NCBI SRA, and these data can be accessed through the BioProject ID: PRJNA342736. All data generated or analysed during this study are included in the source data files.

The following dataset was generated:

| Author(s) | Year | Dataset title | Dataset URL | Database and Identifier |
|---|---|---|---|---|
| Hironobu Fujiwara | 2018 | Transcriptome of hair follicle epidermal stem cells | https://www.ncbi.nlm.nih.gov/bioproject/PRJNA342736/ | NCBI BioProject, PRJNA342736 |

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
