## [Decision Letter]

Thank you for submitting your article "Hair Follicle Epidermal Stem Cells Define a Niche for Tactile Sensation" for consideration by *eLife*. Your article has been reviewed Marianne Bronner as the Senior Editor, a Reviewing Editor, and two reviewers. The reviewers have opted to remain anonymous.

The reviewers have discussed the reviews with one another and the Reviewing Editor has drafted this decision to help you prepare a revised submission.

Summary:

This is an interesting paper that examines the role of compartmentalized epidermal stem cells in tactile sensory neuron formation. The relationship between epidermal stem cells and touch neuron patterning and function is of broad interest to multiple fields. While a lack of mechanistic insights into how EGFL6 regulates lanceolate process morphology and function somewhat diminishes the impact for how epidermal stem cells define a niche for touch sensation, the reviewers were confident that a few additional experiments and changes to the manuscript would make the paper suitable for *eLife*.

Essential revisions:

1) Evidence that αv integrin is the receptor of EGFL6 in lanceolate complexes in vivo is lacking. It is also possible that the αv integrin signals in Figure 2—figure supplement 1E come from the terminal Schwann cells, but the authors did not rule it out. Further, the localization of αv integrins in or near lanceolate complexes doesn't suggest that αv integrins are the receptors of EGFL6 in vivo. Further genetic analyses, such as an analysis of αv integrin conditional knockout mice, would be needed to convincingly show that EGFL6 mediates cell adhesions through αv integrin in vivo. We would encourage this experiment if feasible in a reasonable period of time, but recognize that this could be time consuming in which case a discussion of the limitation of the current results would suffice.

2) Could the hair plucking experiments shown in Figure 4F and 4G also be explained by simply shrinking the volume of the hair follicle after the old hair was plucked? This experiment may not allow one to conclude that hair follicles actively preserve lanceolate complex structures. More experiments would be needed to back up this claim.

3) The reduced α-v integrins upon loss of EGFL6 and observed alteration in lanceolate complexes patterning would suggest possible impairment of cell-cell adhesions. Can the authors increase our understanding of EGFL6 function? Which cell population expresses α-v integrins in-vivo? Are cell-cell adhesions affected in Egfl6-null?

4) The experimental time point for quantification of axon and terminal Schwann cell processes should be indicated. Were all samples collected at the same hair follicle stage? Did the phenotype persist after one additional hair cycle was completed?

5) Can the authors hypothesize why the effect of EGFL6 loss is mild? Is there some compensation by other close homologues (EGFL7, EGFL8, EGFLAM, others…) start being expressed by bulge cells or other neighboring cells?

---

## [Author Response]

[…] Essential revisions:1) Evidence that αv integrin is the receptor of EGFL6 in lanceolate complexes in vivo is lacking. It is also possible that the αv integrin signals in Figure 2—figure supplement 1E come from the terminal Schwann cells, but the authors did not rule it out. Further, the localization of αv integrins in or near lanceolate complexes doesn't suggest that αv integrins are the receptors of EGFL6 in vivo. Further genetic analyses, such as an analysis of αv integrin conditional knockout mice, would be needed to convincingly show that EGFL6 mediates cell adhesions through αv integrin in vivo. We would encourage this experiment if feasible in a reasonable period of time, but recognize that this could be time consuming in which case a discussion of the limitation of the current results would suffice.

We agree that in vivo examination of the contribution of α-v integrin in EGFL6-mediated cell-ECM adhesion is one of the important next steps to extend our findings. We are now starting to obtain α-v integrin floxed mice and several different *CreER* mice to genetically delete α-v integrin specifically in sensory nerves or terminal Schwann cells. These experiments will require more than a year of further work, while they are only made possible by the discoveries described in this paper. So, we also believe that this paper merits publication prior to that considerable extension. In the revised manuscript, we have now included discussion on the limitation of the current results (subsection “Requirement of a Unique ECM in Touch Sensation”), together with the possible contribution of α-v integrin-EGFL6 interactions in lanceolate complex regulation in vivo (subsection “Requirement of a Unique ECM in Touch Sensation”). A detailed examination of α-v integrin tissue localization has been performed (subsection “EGFL6 mediates cell adhesion via αv integrins”) and this is described in our response to comment #3.

2) Could the hair plucking experiments shown in Figure 4F and 4G also be explained by simply shrinking the volume of the hair follicle after the old hair was plucked? This experiment may not allow one to conclude that hair follicles actively preserve lanceolate complex structures. More experiments would be needed to back up this claim.

Thank you for raising this concern. We have now included several experimental data that rule out this possibility. We measured the perimeter of the upper-bulge three days after club hair plucking and found that the perimeter was slightly reduced (Unplucked: 133 ± 2.2 micro meter; Plucked 106 ± 2.2 micro meter in zigzag hair follicles) (Figure 5—figure supplement 1E). However, this reduced perimeter would not be a consequence of global shrinking of the overall perimeter (or volume) of the upper-bulge, like a shrinking balloon, but it could result from specific topological changes in the old bulge, based on the following evidence. (i) The club hair plucking itself did not compromise the two-bulge epidermal architecture (Figure 5—figure supplement 1D). (ii) It has been reported that the hair plucking induces apoptosis in the bulge stem cells within 4.5 hours after hair plucking (Ito et al., 2002; Chen et al., 2015). (iii) Consistent with this, cells with a bright DAPI signal, an indicator of apoptotic event, were observed only in the old bulge of the plucked hair follicles within 4 hours after club hair plucking (Figure 5—figure supplement 1D). (iv) After club hair plucking, the new follicle remained, but the old follicle disappeared (Figure 5F). These results suggest that our club hair plucking procedure specifically depletes the old bulge epidermal cells through inducing apoptosis. Therefore, the club hair plucking does not uniformly reduce overall perimeter (volume) of the bulge, but specifically removes old bulge epidermal cells and changes the topology of the old bulge.

Taking this experimental condition into account, the observed relocation of lanceolate complexes from the caudal side to the rostral side of the hair follicle can be explained as an active relocation of lanceolate complex processes, rather than a passive sliding of a whole array of lanceolate complex processes toward the rostral side. This notion is supported by the fact that, in the plucked follicles, EGFL6 remains at the caudal side even without lanceolate complex processes, while the deposition level of EGFL6 was still low at the rostral side (Figures 5F and G). These results suggest that (i) lanceolate complex processes in the old bulge retracted after club hair plucking, but EGFL6 matrix remained in the caudal side, and (ii) ectopic lanceolate complex processes were formed at the rostral side, but EGFL6 deposition requires additional time as observed in lanceolate complex morphogenesis (Figure 5F and G, Figure 2—figure supplement 1). Therefore, the observed lanceolate complex relocation is not likely to be explained as a consequence of simple shrinking of the volume of hair follicles after the old hair was plucked. We have now described this in Results section.

3) The reduced α v integrins upon loss of EGFL6 and observed alteration in lanceolate complexes patterning would suggest possible impairment of cell-cell adhesions. Can the authors increase our understanding of EGFL6 function? Which cell population expresses α-v integrins in-vivo? Are cell-cell adhesions affected in Egfl6-null?

Thanks to these questions, we have now examined the detailed tissue localization of α-v integrins. Alpha-v integrins were localized to the axonal endings and distributed on their collar matrix side, where EGFL6 was deposited, but they did not colocalize with terminal Schwann cell processes (Figure 3H). This tissue distribution was confirmed by three distinct α-v integrin antibodies (see key resources table). We have also examined the localization of other integrins, including beta1, alpha3 and alpha6 integrins, and they were also detected predominantly at the outer or inner surfaces of the lanceolate complex endings, where the collar matrix and basement membrane exist (Figure 3H). These results indicate that the EGFL6-α-v integrin interactions are restricted at the collar matrix side of axonal endings and involved in axon–ECM interactions, but not directly involved in cell–cell adhesions between axonal endings and terminal Schwann cell processes.

Nevertheless, we have examined defects in cell–cell adhesions that could be indirectly caused by *Egfl6* deletion. Since molecules mediating cell–cell adhesions between axonal endings and terminal Schwann cells remain unknown, we examined the ultrastructural morphology of cell–cell adhesion sites in electron microscopic images of lanceolate complexes in *Egfl6* knockout mice. Although we found lanceolate complex endings that show imperfectly sandwiched morphologies in the KO mice, we did not detect obvious abnormalities in the morphology of cell–cell interfaces and cell–cell adhesion structures (Figure 4—figure supplement 2). We describe these results in the revised manuscript (see Results section and Discussion section). Since receptor information of EGFL6 is one of the major issues in this revision and would be of interest to readers, we decided to move this part from a supplemental figure to a main figure (new Figure 3).

4) The experimental time point for quantification of axon and terminal Schwann cell processes should be indicated. Were all samples collected at the same hair follicle stage? Did the phenotype persist after one additional hair cycle was completed?

Quantification of axon and terminal Schwann cell processes were performed with 8-week-old *Egfl6* KO/*Ret*-eGFP mice (Figure 4A–C, Figure 4—figure supplement 1A–D). Thus, all samples for this quantification were collected at the same hair follicle stage, namely the second telogen. This information was not stated in the figure legend of Figure 4—figure supplement 1A–D. We apologize for the lack of detail and have now added this in the revised manuscript.

To examine whether the lanceolate complex phenotypes persist after one additional hair cycle was completed, we investigated the lanceolate complex structure of *Egfl6* knockout mice at the first telogen (P19), rather than at the third telogen. To obtain third telogen skin samples, three months of rearing period of the mutant mice was needed, which would not allow us to complete the requested experiment within the revision period. We believe that examining the first and second telogen hair follicles allows us to address this question. Quantitative 3D histomorphometric analysis of nerve and terminal Schwann cell processes revealed occasional appearance of disorganized parallel axonal endings and terminal Schwann cell processes, together with increased number of axon overlaps (Figure 4—figure supplement 3, subsection “EGFL6 is required for the proper patterning and touch responses of”). Thus, disorganized lanceolate complex phenotypes persisted after an additional hair cycle was completed.

5) Can the authors hypothesize why the effect of EGFL6 loss is mild? Is there some compensation by other close homologues (EGFL7, EGFL8, EGFLAM, others…) start being expressed by bulge cells or other neighboring cells?

We have examined the mRNA expression levels of all 11 EGFL family member genes (*Egfl2, Egfl3, Egfl4, Egfl5, Egfl6, Egfl7, Egfl8, Egfl9, Egflam, Megf10* and *Npnt*) in CD34^+^epidermal stem cells, which contain mid-bulge epidermal stem cells and a part of *Gli1*^+^ upper-bulge epidermal stem cells, from wild-type and *Egfl6* knockout mice. The expression of 7 family members, including *Egfl6*, was detected in wild-type (Figure 5—figure supplement 2). But no significant difference in the expression levels of these genes was observed between wild-type and mutant (except for mutated *Egfl6*). The following genes were under a detectable level in both genotypes; *Egfl3, Egfl8, Egfl9, Megf10*. The EGFL family members expressed in the bulge region could functionally compensate for the deletion of EGFL6 even without changing their expression levels.

We have also examined the expression of several integrins, such as integrins beta1, alpha3 and alpha6. They are also expressed in the axonal endings and accumulated to the axon-collar matrix interfaces (Figure 3H). There are many evidences of functional redundancy and compensation in cell-ECM interactions by other family members of integrins and ECM molecules in vivo (Lowell and Mayadas, 2012). Thus, compensation by other cell-ECM interactions in lanceolate complexes could account for the mild phenotype of *Egfl6* mutants. Further analysis on upper-bulge–specific ECM molecules and their receptors will provide insights into how tactile sensation is regulated by the ECM. We have now described this in the Discussion section.